# Structural analysis of 3'UTRs in insect flaviviruses reveals novel determinants of sfRNA biogenesis and provides new insights into flavivirus evolution

Andrii Slonchak [1✉], Rhys Parry [1], Brody Pullinger[1], Julian D. J. Sng [1], Xiaohui Wang[1], Teresa F. Buck[1,4], Francisco J. Torres [1], Jessica J. Harrison[1], Agathe M. G. Colmant [1], Jody Hobson-Peters [1,2], Roy A. Hall[1,2], Andrew Tuplin[3] & Alexander A. Khromykh [1,2✉]

Subgenomic flaviviral RNAs (sfRNAs) are virus-derived noncoding RNAs produced by pathogenic mosquito-borne flaviviruses (MBF) to counteract the host antiviral response. To date, the ability of non-pathogenic flaviviruses to produce and utilise sfRNAs remains largely unexplored, and it is unclear what role XRN1 resistance plays in flavivirus evolution and host adaptation. Herein the production of sfRNAs by several insect-specific flaviviruses (ISFs) that replicate exclusively in mosquitoes is shown, and the secondary structures of their complete 3'UTRs are determined. The xrRNAs responsible for the biogenesis of ISF sfRNAs are also identified, and the role of these sfRNAs in virus replication is demonstrated. We demonstrate that 3'UTRs of all classical ISFs, except *Anopheles spp*-asscoaited viruses, and of the dual-host associated ISF Binjari virus contain duplicated xrRNAs. We also reveal novel structural elements in the 3'UTRs of dual host-associated and *Anopheles*-associated classical ISFs. Structure-based phylogenetic analysis demonstrates that xrRNAs identified in *Anopheles spp*-associated ISF are likely ancestral to xrRNAs of ISFs and MBFs. In addition, our data provide evidence that duplicated xrRNAs are selected in the evolution of flaviviruses to provide functional redundancy, which preserves the production of sfRNAs if one of the structures is disabled by mutations or misfolding.

[1] School of Chemistry and Molecular Biosciences, University of Queensland, Brisbane, QLD, Australia. [2] Australian Infectious Diseases Research Centre, Global Virus Network Centre of Excellence, Brisbane, QLD, Australia. [3] School of Molecular and Cellular Biology, University of Leeds, Leeds, UK. [4]Present address: Institute for Medical and Marine Biotechnology, University of Lübeck, Lübeck, Germany. ✉email: a.slonchak@uq.edu.au; a.khromykh@uq.edu.au

Flaviviruses represent a diverse group of positive-strand RNA viruses, including arthropod-borne human pathogens such as Zika, Dengue and West Nile virus, and a plethora of non-pathogenic viruses. The *Flavivirus* genus can be divided into the following ecological groups: mosquito-borne flaviviruses (MBFs), which circulate between mosquito and vertebrate (avian, equine or human) hosts; tick-borne flaviviruses (TBFs) that are maintained in nature in tick-vertebrate cycle[1]; viruses that only infect vertebrates and are thought to be transmitted horizontally[2] (no known vector flaviviruses, NKVFs) and insect-specific flaviviruses (ISFs) that infect mosquitoes or sand flies and are believed to circulate predominantly via vertical transmission[3]. To date, pathogenic dual-host viruses (mosquito- and tick-borne flaviviruses) are the most well-studied flaviviruses. To replicate in diverse hosts, these viruses have evolved multiple mechanisms to counteract, subvert and evade antiviral responses of arthropods and vertebrates. One of them is by producing noncoding viral RNA named subgenomic flaviviral RNA (sfRNA) (reviewed in[4]). In infected cells, viral genomic RNA is subjected to degradation by the host 5'-3' exoribonuclease XRN1. XRN1 is a highly processive enzyme with a helicase activity, which can unwind and fully digest virtually any RNA[5]. However, flaviviruses contain uniquely folded RNA elements in their 3'UTRs that can halt the progression of XRN1[6]. Stalling of XRN1 prevents complete degradation of viral genomic RNA and results in accumulation of 3'UTR-derived sfRNAs in the infected cells[7] (Supplementary Fig. 1A). Multiple studies have demonstrated that sfRNAs facilitate viral replication and pathogenesis by inhibiting IFN response in vertebrates[8–12] and RNAi[13,14] or apoptosis[15] in mosquitoes.

XRN1-resistant elements are believed to have conserved secondary and tertiary structures (Supplementary Fig. 1B, C) within each clade of flaviviruses[4]. Commonly, all flavivirus xrRNAs characterized to date are formed by the stem-loops (SL) that contain a three-way junction between three RNA helices (P1, P2 and P3) and a pseudoknot (PK) formed by the terminal loop (L2) of P2 helix (Supplementary Fig. 1C). In addition, xrRNAs contain a small pseudoknot (sPK) within a junction between P1 and P3 helices (Supplementary Fig, 1C). The pseudoknots and noncanonical interactions within xrRNAs determine their unique tertiary conformation in which the 5'-end of the RNA passes through a ring-like structure[16,17] (Supplementary Fig. 1B). This unique fold determines the resistance of xrRNAs to XRN1 as the RNA ring creates a roadblock for the progression of the enzyme[18] (Supplementary Fig. 1A). Secondary structures have been experimentally determined for xrRNAs of mosquito-borne[6,19,20], tick-borne[21], no known vector flaviviruses[21,22] and an insect specific Cell fusing agent virus (CFAV)[21]. In addition, crystal structures have been solved for three mosquito-borne flavivirus xrRNAs[16,17] and xrRNA of no known vector flavivirus Tamana Bat virus (TABV)[22]. Based on the conserved structural elements, xrRNAs are classified into two classes – class 1 xrRNAs present in mosquito-borne flaviviruses and class 2 xrRNAs present in tick-borne and no known vector viruses[21]. Within class 1 xrRNAs, two additional subclasses can be distinguished – subclass 1a includes the typical mosquito borne flavivirus xrRNAs that contain 5-nt P1 helix, 6-nt pseudoknot and conserved unpaired nucleotide between P2 and P3[16] (Supplementary Fig. 1C). Subclass 1b xrRNAs are identified in no known vector flavivirus Tamana Bat virus (TABV) and contain 3-4-nt P1 helix, often with noncanonical A-C base pairing at the base of the helix, 3-nt pseudoknot and no unpaired bases between P2 and P3[22,23] (Supplementary Fig. 1C).

3'UTRs of mosquito borne flaviviruses typically contain duplicated stem-loop (SL) based xrRNAs followed by two dumbbell (DB) structures and a 3'-terminal stem-loop (3'SL)[24]. The duplicated stem-loops give rise to two sfRNAs of different length[25]. In addition, two smaller sfRNAs are produced by mosquito borne flaviviruses with their 5' ends likely located at the beginning of dumbbell structures. However, the biogenesis mechanism of these RNAs is unclear as dumbbells have been recently shown to lack XRN1 resistance[26]. Currently, it is not fully understood why MBFs acquired additional xrRNAs and whether sfRNA isoforms of different lengths have redundant or specialized functions. One study demonstrated that Dengue virus 2 (DENV2) accumulates mutations in the individual xrRNAs and switches between the production of longer and shorter sfRNAs when switching hosts[10]. This suggests that duplication of the structural elements within 3'UTR represents an adaptation to the dual host life cycle, and individual sfRNA species are specifically adapted to function in different hosts. However, the apparent functional specialization of sfRNA isoforms was only shown for DENV2, and the hypothesis about structure duplication being an adaptation to host switching has not been unambiguously proven or disproven.

Despite the extensive knowledge about sfRNAs accumulated to date, several questions regarding their biogenesis, functions, and role in flavivirus evolution still remain unanswered. In particular, it is unknown how the structural diversity of XRN1-resistant elements have evolved and whether the different types of xrRNAs appeared independently in different flaviviruses or diverged from a common ancestor. It is also unclear whether duplicated xrRNAs are unique to mosquito born flaviviruses or present in the viruses with a single-host restricted life cycle. In addition, sfRNAs have been primarily studied in pathogenic flaviviruses, while their biogenesis and functions in non-pathogenic flaviviruses remain largely unknown. For instance, the production of sfRNA has so far been shown only for one insect flavivirus – cell fusing agent virus (CFAV)[21].

Herein we aimed to elucidate the biogenesis and functions of sfRNAs in insect-specific flaviviruses. ISFs include two phylogenetically distinct lineages – (i) lineage I or classical ISFs (cISFs) that independently diverged from an ancient flavivirus ancestor and (ii) lineage II or dual host-associated ISFs (dISFs) that are believed to evolve from mosquito-borne flaviviruses that lost the ability to infect vertebrates[3,27]. Identifying xrRNA structures responsible for the production of sfRNAs in representatives of both ISF lineages and demonstrating the functional significance of ISF sfRNAs should allow the elucidation of whether production of sfRNAs contributes to host adaptation and unravel how XRN1 resistance has evolved in flaviviruses. Two previous attempts to predict the structure of cISF 3'UTRs did not reveal high-order secondary structures, identifying only short inverted repeats[24,28]. However, more recent work suggested that dISFs may contain class 1b xrRNAs, similar to those identified in TBAV[23]. While the 3'UTRs of dISF were predicted to contain a single SL and a single dumbbell followed by 3'SL[24], the role of these predicted structures in XRN1 resistance and production of sfRNAs by dISFs have not been experimentally demonstrated.

Herein we demonstrate that dISFs and cISFs, including the most phylogenetically distant *Anopheles*-associated flaviviruses, generate sfRNAs by employing XRN1-resistant structured RNA elements. Using chemical probing for RNA structure, we experimentally determine the secondary structures of the complete 3'UTRs of classical and dual host-associated ISFs. We discover that dISFs contain a novel pseudoknot element that increases XRN1-resistance of class 1a xrRNAs for the robust production of sfRNAs. We also find that cISFs and the dISF BinJV contain multiple copies of xrRNAs. However, *Anopheles*-associated cISFs contain only a single XRN1-resistant structure, which has features of both class 1a and class 1b structures and is therefore ancestral to xrRNAs of insect-specific and mosquito-borne flaviviruses. Furthermore, our sequence- and structure-

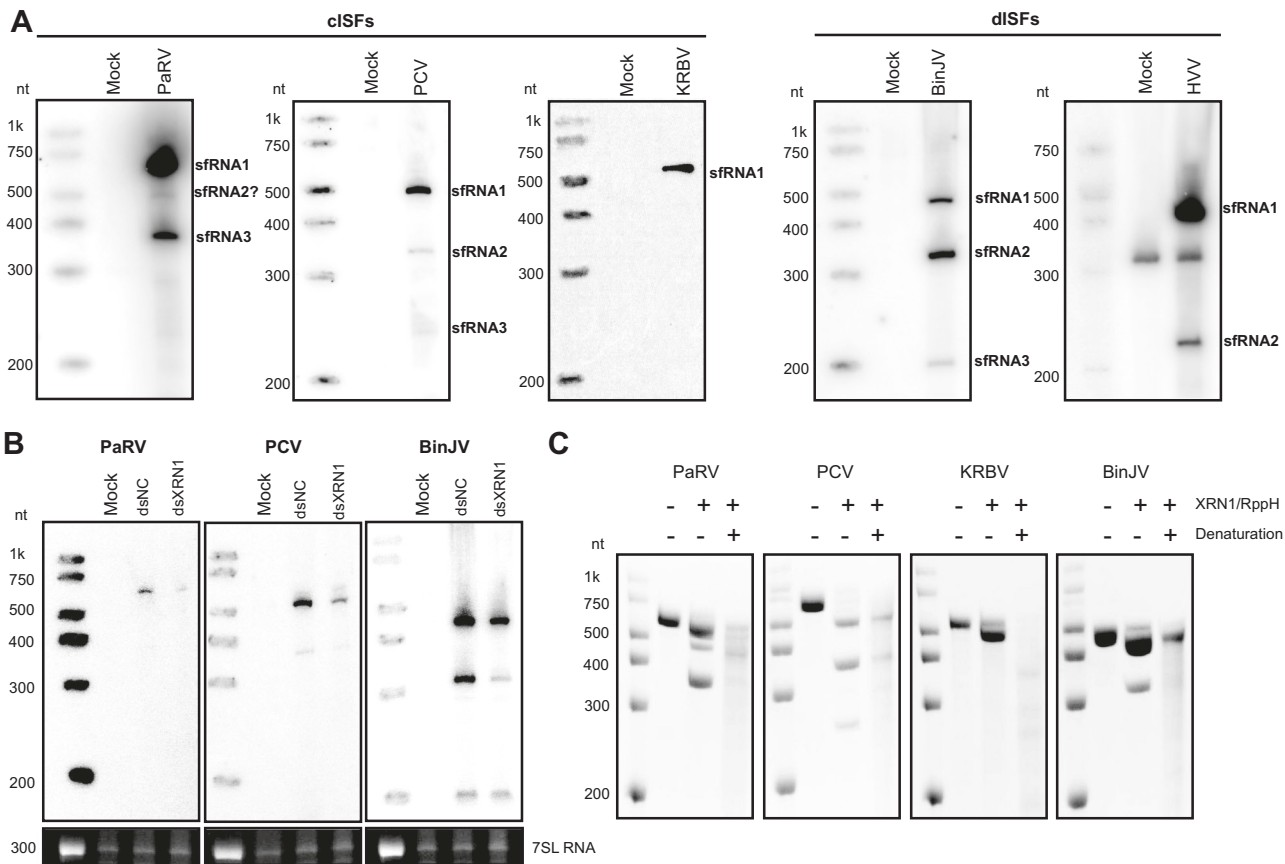

**Fig. 1 Classical and dual host-associated ISFs produce sfRNAs by employing XRN1-resistance mechanism. A** Northern blot detection of sfRNAs produced by ISFs. For PaRV, PCV, BinJV and HVV C6/36 cells were infected at MOI = 1. Total RNA was isolated at 5 dpi. For KRBV, total RNA was isolated from virus-positive and virus-negative (Mock) *Anopheles* mosquitoes. RNA was then used for Northern blotting with the probe complementary to the last 25nt of viral 3'UTRs. **B** The effect of XRN1 knock-down on the production of sfRNAs by ISFs. Aag2 cells were transfected with dsRNA against *Aedes aegypti* XRN1 (dsXRN1) or GFP (dsNC) and infected with respective viruses at MOI = 1 at 24hpt. At 48hpi, total RNA was isolated from the cells and used for Northern blotting as in (**A**). Bottom panels represent the Et-Br staining of the gels used for Northern transfer with 7SL cellular RNA visualised as a loading control. **C** In vitro XRN1 resistance assay with ISF 3'UTRs. RNA corresponding to 3'UTRs of ISFs was transcribed in vitro, briefly heated and then refolded by gradual cooling to 28 °C or placed on ice to preserve the denatured state. Samples were then treated with purified XRN1 and RppH (to convert 5'PPP into 5'P) and analysed by electrophoresis in denaturing PAAG. Gels were stained with ethidium bromide (Et-Br). All images are representative of at least two independent experiments that produced similar results.

based phylogenetic analysis of 3'UTRs reveals that the evolution of flaviviruses involves several xrRNA duplication events that occur independently in cISF, MBF and dISF clades, indicating their significant benefit for viral fitness. Finally, we use ISF mutants with impaired functions of individual or all xrRNAs to demonstrate that different sfRNA species likely have redundant functions in ISFs. Our data thus indicate that duplication of xrRNAs and the acquisition of novel structural elements that increase XRN1-resistance of xrRNAs is an evolutionarily conserved mechanism to ensure robust sfRNA production and protection from potentially damaging impacts of inadvertent mutations.

## Results

**Classical and dual host-associated ISFs produce sfRNAs via XRN1 resistance mechanism.** To determine whether phylogenetically divergent classical and dual host-associated ISFs are capable of sfRNA production, we performed a Northern blot of total RNA isolated from C6/36 cells infected with *Aedes*-associated cISF Parramatta River Virus (PaRV)[29], *Culex*-associated cISF Palm Creek virus (PCV)[30] and dISFs Binjari virus (BinJV)[31,32] and Hidden valley virus (HVV)[32]. To test for production of sfRNA by *Anopheles*-associated Karumba virus

(KRBV), which is unable to replicate in all tested laboratory cell lines[33], RNA from virus-positive field-collected mosquitoes was used. All tested viruses were found to produce sfRNAs (Fig. 1A). PCV and BinJV produced three sfRNAs of different lengths. In contrast, HVV produced two sfRNAs, PaRV produced two major sfRNAs species and potentially one less abundant sfRNA, while *Anopheles*-associated flavivirus KRBV produced only one sfRNA (Fig. 1A). In infection with the viruses that produce multiple sfRNAs, the largest sfRNA isoform was generally the most abundant, while apart from PaRV, the smallest sfRNA was produced by all tested viruses in the least quantity (Supplementary Fig. 2A). To further identify the structural determinants of sfRNA biogenesis in ISFs, we determined the 5'-ends of identified sfRNAs using RNA ligation-mediated RT-PCR sequencing method[10] (see Methods). We were able to identify 5'-ends of PaRV, PCV and BinJV sfRNAs and map them to viral 3'UTRs (Supplementary Fig. 3). We did not obtain an amplification product for KRBV sfRNA, which was likely due to the low viral load in the mosquito samples.

To elucidate if XRN1 is responsible for sfRNA biogenesis in ISFs, the effect of XRN1 knock-down on the production of sfRNA by PaRV, PCV and BinJV were assessed. Transfection of dsRNA against XRN1 resulted in 90–98% depletion of XRN1 mRNA

through the time course of the experiment (Supplementary Fig. 2B) and reduced production of sfRNAs by PaRV and PCV by approximately 50% (Fig. 1B, Supplementary Fig. 2C). It also resulted in decreased accumulation of BinJV sfRNA-1 and -2 by 50% and 80%, respectively (Fig. 1B, Supplementary Fig. 2C). These results show that sfRNAs in cISF and major sfRNA species in dISFs are produced via incomplete digestion of 3′UTRs by XRN1. To assess if XRN1 resistance in ISFs is determined by structured RNA elements, the ability of XRN1 to digest folded and denatured viral 3′UTRs into sfRNA in vitro was tested. Treatment with XRN1 resulted in partial degradation of refolded in vitro transcribed 3′UTRs of PaRV, PCV, BinJV and KRBV (Fig. 1C) and production of sfRNAs that had the same length as those detected in infection (Fig. 1A). However, 3′UTRs of all examined viruses became highly susceptible to complete degradation by XRN1 if RNA was denatured (Fig. 1C). This indicates that XRN1 resistance in cISFs and dISFs is determined by structured RNA. The in vitro XRN1 digestion assay also shows that XRN1 is responsible for the production of sfRNA by KRBV, which we could not use in the knock-down experiment due to the inability of this virus to replicate in cell culture. Notably, sfRNA-3 of BinJV was not produced in in vitro digestion reaction (Fig. 1C), and its level in virus-infected cells was not affected by XRN1 knock down (Fig. 1B, Supplementary Fig. 2C). These results support previous findings of XRN1-independent production of sfRNAs from the flavivirus dumbbell structures[26].

In conclusion, we demonstrated that cISFs and dISFs produce sfRNAs due to the presence of structured XRN1-resistant RNA elements in their 3′UTRs. We also found that all tested ISFs except KRBV produce multiple sfRNA species, which infers the existence of several XRN1 resistant structures in their 3 UTRs. In addition, we identified 5′-ends of PaRV, PCV, and BinJV sfRNAs for further mapping of the structured elements in viral 3′UTRs (putative xrRNAs).

**Dual host-associated ISFs contain novel structural elements that assist canonical xrRNAs in the formation of XRN1-resistant structures**. To determine the secondary structure of BinJV 3′UTR and identify structural determinants of sfRNA biogenesis in dISFs, we employed selective 2′-hydroxyl acylation analysed by primer extension (SHAPE). This method identifies paired and unpaired nucleotides in the in vitro transcribed and refolded RNA[34]. The base pairing information is then used to guide the computational folding of RNA structures. SHAPE results demonstrated that the BinJV 3′UTR contains two pseudoknot-forming stem-loops with 3-way junctions, a single copy of dumbbell element and a canonical 3′SL with a preceding short hairpin (Fig. 2A). PK-forming stem-loop elements of BinJV had features of class 1a xrRNAs such as unpaired C nucleotide in P2-P3 junction, conserved GU upstream of P1 stem that formed a small pseudoknot with nucleotides between P3 and P1, and 5-6nt long-range pseudoknot formed by the L2 loop (Fig. 2B, C). Therefore, they structurally fit into class 1a of xrRNAs typical for MBFs and have significant sequence and structural homology with them (Fig. 2C). Looking at BinJV xrRNAs (Fig. 2B), we expect a U-A-U base triple between U4 and A22-U43 (the numbers are for xrRNA1) (Fig. 2C), and the base-pairs between A2-G3 and C42-U43 into the three-way junction (sPK Fig. 2B, C). Therefore, we can be confident that a ring of 14 nt will form, and this almost certainly will be when the top of P3 rearranges to cause A38 to extrude out from the helix and interact with U52 in a long-range reverse-Watson-Crick pair. These structural features were seen in the Zika xrRNA[16]. The homology between the Zika sequence and secondary structure and the BinJV sequence and secondary structure (Fig. 2C) means that both BinJV stem loops

can be confidently predicted to fold almost identical to the Zika xrRNA1. Therefore, we can say with confidence that they form the ring-like structure required for XRN1 resistance. Moreover, BinJV SLs are located at positions corresponding to the 5′-ends of identified sfRNAs (Supplementary figure 3, Fig. 2A). In addition, disruption of pseudoknot interactions in SLI and SLII by mutations greatly reduced the production of sfRNA1 and sfRNA2, respectively, in in vitro XRN1 digestion assay (Fig. 2D). This directly confirmed the XRN1 resistant nature of these elements and the requirement of pseudoknots for their resistance. Therefore, we concluded that BinJV contains two copies of xrRNAs.

Further examination of the structure of the BinJV 3′UTR revealed that small stem-loops CS3 and RCS3 downstream of xrRNA1 and xrRNA2 also formed pseudoknots that we named novel pseudoknots nPK1 and nPK2 (Fig. 1A). Although similar short stem-loops are present in the genomes of all mosquito-borne flaviviruses, they are not involved in pseudoknot interactions in any of the viruses for which structural data on the 3′UTR is available[6,19] and have not been predicted to form PKs. Interestingly, in contrast to other flaviviruses[6,7,19,21], disruption of PK1 in xrRNA1 or PK2 in XRNA2 of BinJV by mutations did not completely abolish XRN1-resistance (Fig. 2D). Hence, we hypothesized that nPKs might stabilise xrRNAs or confer additional XRN1 resistance. To test this hypothesis, we assessed the effects of mutations that disrupt nucleotide pairing in nPK1 and nPK2 on the production of BinJV sfRNAs in vitro. Notably, disruption of nPK1 significantly reduced the generation of sfRNA-1 (Fig. 2D, E). We also observed decreased production of sfRNA2 after mutation in nPK2, although the difference was not statistically significant. This, as well as the complete disappearance of sfRNA-2 upon PK2 disruption, was likely due to the initially very low level of this sfRNA produced from the WT sequence in vitro (Fig. 2D, E). These results indicate that nPKs contribute to the production of corresponding sfRNAs from BinJV 3′UTR. To determine whether the small stem-loops with novel pseudoknots are XRN1-resistant on their own, we tested their ability to protect otherwise XRN1-sensitive fragment of GFP RNA from degradation by XRN1 in vitro (Supplementary figure 4A, B). The XRN1 resistance assay demonstrated that, as expected, insertion of the canonical stem-loop of BinJV xrRNA1 into the GFP RNA fragment efficiently stalled XRN1 and produced RNA fragment of smaller length when exposed to the XRN1 activity in vitro (Supplementary Fig. 4B). In contrast, the GFP RNA fragment with insertions of novel PK structures nPK1 or nPK2 alone were completely degraded by XRN1, similar to the RNA fragment, which contained scrambled xrRNA1 insertion (Supplementary figure 4B). These results indicate that novel pseudoknots of BinJV 3′UTR are not XRN1-resistant without the surrounding 3′UTR context. Instead, they confer an additional XRN1 resistance to the upstream canonical xrRNAs and allow them to stall XRN1 progression even if the mutations disrupt their main pseudoknots.

To elucidate whether novel pseudoknots (nPKs) are unique for BinJV or represent a more common feature for all dISFs, we first used IPknot[35] software to predict pseudoknot interactions in 3′UTRs of all dual host-associated ISFs for which complete 3′UTR sequences were available. The analysis demonstrated that all dISFs (Supplementary Fig. 5A) except NHUV (Supplementary figure 5B) contain a putative pseudoknot formed by small stem-loops downstream of the canonical xrRNAs. We then performed structure-based sequence alignment of dISF xrRNAs-nPK regions (Supplementary Fig. 5C), built a covariance model[36] and used it to identify the consensus secondary structure (Fig. 2F). The covariance analysis demonstrated that canonical xrRNA stem loop-PKs had high sequence and structure homologies with two conserved and at least three covarying nucleotide pairs (preserve

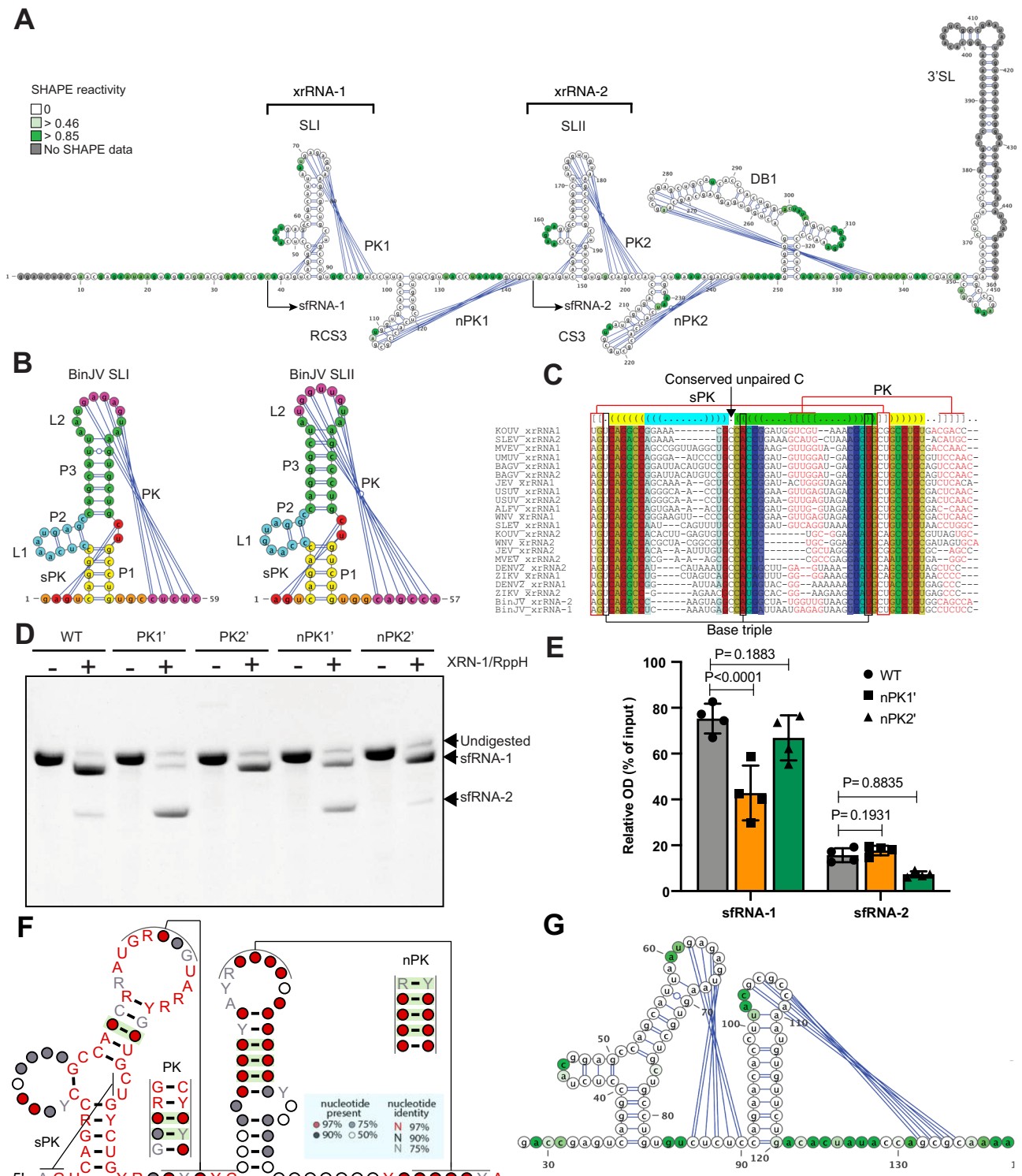

**Fig. 2 dISFs contain novel SL-PK elements in addition to canonical class 1a xrRNAs. A** Secondary structure of BinJV 3'UTR generated by SHAPE. SL – stem-loop, DB – dumbbell, RCS3 – reverse conserved sequence 3, CS3 – conserved sequence 3, PK – pseudoknot. **B** Secondary structures of BinJV stem-loops. **C** Structure-based sequence alignment between BinJV and MBF xrRNAs. **D** XRN1 resistance assay with WT and mutated BinJV 3'UTRs. Mutations PK1' (GAGAG- > CUCUC), PK2' (UGGUUG- > ACCAAC), nPK1' (UAGCG- > AUCGC) and nPK2' (GCGUC- > CGCAG) were introduced into the terminal loop regions of the corresponding stem-loops. The image is representative of four independent experiments that produced similar results. **E** Densitometry analysis of (**D**). The values are the means of four independent experiments ± SD. Statistical analysis is two-sided one-way ANOVA. **F** Consensus structure of dISF xrRNAs built based on the covariance model. Covarying base pairs are highlighted in green (**G**) Secondary structure of HVV xrRNA generated by SHAPE.

complementary if vary) in the PK region (Fig. 2F). The smaller stem-loops were found to lack sequence conservancy but maintained a conserved structure with at least three covarying nucleotides in the stem. Importantly, novel PKs were shown to be structurally conserved with four highly (>97%) covarying nucleotide pairs. This indicates that the ability to form novel pseudoknots is conserved among dISFs. To validate the predicted nPKs in other dISF representative, we performed SHAPE on the fragment of the 3'UTR of Hidden Valley Virus (HVV), which demonstrated nucleotide pairing between the loop region of the hairpin element and a downstream sequence (Fig. 2G), confirming that novel pseudoknots are not unique to BinJV. The SHAPE data also further supported the consensus structure of the canonical xrRNAs in dISFs (Fig. 2G) generated based on the covariance model.

Therefore, we showed that dISFs have a novel sub-class of xrRNAs containing additional structural elements formed by the small stem-loop and pseudoknot downstream of the canonical xrRNA structure. In the current classification of xrRNA types, they would be class 1c xrRNAs. They occur commonly in dISFs, but not in MBFs and employ novel pseudoknot interactions to confer an additional XRN1 resistance to the upstream canonical xrRNA structure for the production of sfRNAs. In addition, the experimental identification of the secondary structure of dISF 3'UTR demonstrated that this group of flaviviruses can contain duplicated SLs.

**Classical ISFs PaRV and PCV contain multiple copies of class 1b xrRNAs.** To identify structural determinants of sfRNA biogenesis in classical ISFs, we determined the secondary structure of 3'UTRs of PaRV and PCV using SHAPE. SHAPE-assisted folding of PaRV 3'UTR revealed that it contains five stem-loop elements, four of which form pseudoknots and are followed by one or two small stem-loops similar to CS3 and RCS3 elements of MBFs/dISFs (Fig. 3A). PaRV 3'UTR also contained a 3'-terminal SL similar to all other flaviviruses while having no dumbbell elements (Fig. 3A). PK-forming stem-loops in PaRV 3'UTR had properties of class 1b xrRNAs and generally corresponded to the recently predicted structures[23] with minor differences. Two of them (SLI and SLIII) were mapped to the 5'-termini of the PaRV sfRNA-1 and sfRNA-3 (Fig. 3A). Although we were unable to determine the exact 5'-end of less abundant sfRNA-2, the length of this RNA infers that it should start at SLII in PaRV 3'UTR (Figs. 1A, 3A). Notably, neither Northern blot of RNA from infected cells (Fig. 1A) nor in vitro XRN1 digestion (Fig. 1C) revealed production of sfRNA from SLIV, suggesting that SLIV does not represent a functional xrRNA despite the apparent structural similarities. Comparing structures of PaRV SLI, II, and III (Fig. 1B) revealed that xrRNA2 acquired a mutation of U to A at the first position of L2, which created additional base pair in the P3 helix, extending its length. Given that the P3 helix is involved in forming the RNA ring, this should result in the structure that is more topologically relaxed than the conformation of xrRNA1 and xrRNA3 and thus likely to be more prone to digestion by XRN1. This explains why PaRV xrRNA2 has poor XRN1 resistance and produces only a fractional amount of sfRNA2 (Fig. 1A, C). To further validate the identified xrRNAs, we assessed the effect of PK disruption on the generation of PaRV sfRNAs in vitro. Mutations that disrupt base pairing in PKs of xrRNA1, xrRNA2 and xrRNA3 completely abolished XRN1 resistance in the corresponding regions of PaRV 3'UTR and prevented the generation of sfRNA-1, sfRNA-2, and sfRNA-3, respectively (Fig. 3C). This confirms that PK-forming stem-loops SLI-III represents structural determinants of sfRNA biogenesis in PaRV 3'UTR. Finally, we queried the origin of the multiple

xrRNAs in PaRV. The structure-based sequence alignment revealed a high level of homology between PaRV xrRNAs, with xrRNA-2 being the most divergent (Fig. 3D). This indicates that multiple copies of PaRV SLs likely appeared due to duplication events with subsequent accumulation of mutations in xrRNA2, which decreased XRN1 resistance.

Based on SHAPE-guided folding of PCV 3'UTR we identified three copies of the structural element, which have the features of potential xrRNAs. Each of them consisted of a PK-forming SL followed by two smaller SLs. At the 3'-end PCV 3'UTR contained two short hairpins and the 3'SL (Fig. 4A). The PK-forming SLs of PCV have features of class 1b xrRNAs, such as 3nt P1 stem, 3nt long-range PK formed by L2 loop and an internal PK formed by nucleotides CGG/A located between P3 and P1 (Fig. 4A, B). In addition, SLII and SLIII of PCV contained a noncanonical A-C pairing in the base of the P1 stem, which is a distinctive feature of class 1b xrRNAs. The SLI did not involve noncanonical base pairing and had a very long P2 helix, which hasn't been previously observed in flavivirus xrRNAs (Fig. 4A, B). Each of the identified SLs mapped accurately to the 5'-ends of PCV sfRNAs (Supplementary Fig. 3). Together with the structural features of SLs, this allowed us to identify them as putative xrRNAs. Their XRN1 resistance was further confirmed by in vitro XRN1 digestion assay, which showed that mutations in the L2 loop of SLI, SLII and SLIII preventing the formation of PK interactions abolished production of sfRNA-1, 2 and 3, respectively (Fig. 4C). Structure and sequence alignment demonstrated that despite the apparent structural difference of SLI, all three xrRNAs of PCV had a high degree of homology in all their structural elements except for the P2 helix and thus likely resulted from a duplication of a single ancestral structure (Fig. 4D).

In conclusion, we determined secondary structures of the 3'UTRs of classical ISFs. We demonstrated that 3'UTRs of PaRV and PCV contain multiple copies of class 1b xrRNAs and a conserved 3'SL. In addition, PaRV was found to contain other xrRNA-like elements that could represent xrRNAs that lost their functions due to the accumulation of mutations. PARV 3'UTR also has xrRNA (xrRNA2) that partially lost its XRN1 resistance due to a mutation. We also obtained the evidence that multiple xrRNAs in the 3'UTRs of PaRV and PCV emerged due to duplication, which happened early enough in their evolution to allow accumulation of sequence difference between xrRNAs of each virus while maintaining structural conservancy.

***Anopheles*-associated ISFs contains a single copy of xrRNA, which exhibits characteristics of class 1a and class 1b xrRNAs.** Recently discovered *Anopheles*-associated classical ISFs are the most phylogenetically divergent group of insect-specific flaviviruses that were suggested to represent the closest clade to the common ancestors of ISFs and MBFs[33]. In this study, we used Karumba virus (KRBV) to represent this clade and showed that this virus appears to be the only flavivirus analyzed so far that produced only one sfRNA species (Fig. 1A, C). SHAPE analysis of KRBV 3'UTR demonstrated that it does not contain a typical 3'SL element at the 3'end, which was believed to be conserved in all flaviviruses (Fig. 5A). Instead, it has a 3'-terminal stretch of unstructured RNA preceded by a large stem-loop with a three-way junction (SLII in Fig. 5A), which structurally resembles xrRNAs of TBFs and NKVFs[21]. In addition, SHAPE indicated that nucleotides in the terminal loop of this structure are likely paired, suggesting a potential PK between U429-A432 and U476-A480 (Fig. 5A). However, we cannot confidently say whether this PK exists as the U476-A480 region was located within the SHAPE primer and did not produce structural data. In any case, this structure was unlikely to be responsible for the biogenesis of

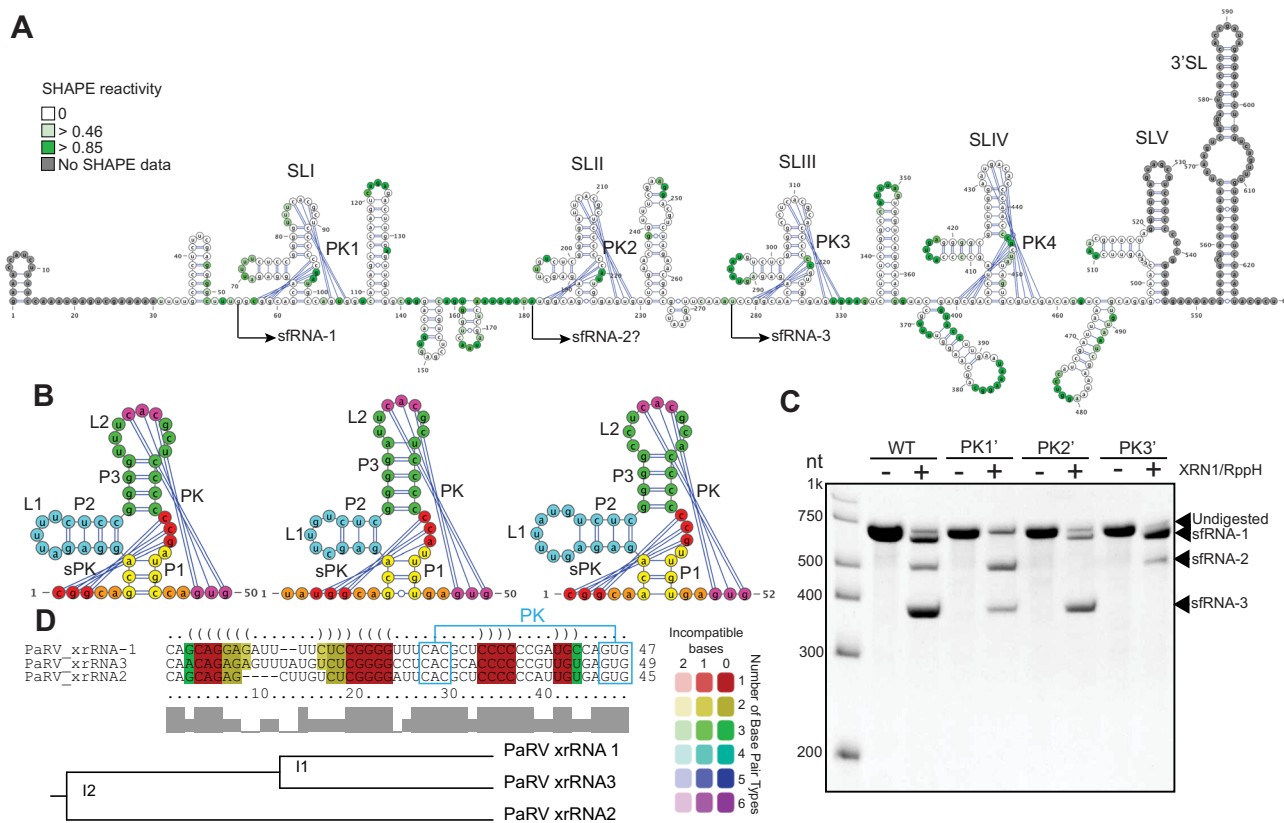

**Fig. 3 cISFs PaRV contains multiple copies of divergent class 1b xrRNAs. A** Secondary structure of PaRV 3'UTR generated using SHAPE-assisted folding. Colour intensity indicates NMIA reactivity. Values are the means from 3 independent experiments, each with technical triplicates. SL – stem loop, PK -pseudoknot. **B** Secondary structure of PaRV xrRNAs. Each structural element is shown in colour. **C** In vitro XRN1 resistance assay with WT and mutated PaRV 3'UTRs. PK1', PK2' and PK3' mutations were introduced into the terminal loop regions of SLI, SLII and SLIII, respectively, and represented the CAC - > GUG change in the PK-forming regions. Image is representative from three independent experiments that showed similar results. **D** Structure-based alignment of PaRV xrRNAs was performed using LocARNA.

KRBV sfRNA as the location of this SL did not match the observed sfRNA length (Fig. 1A, C).

Another structured region was identified at the 5' end of KRBV 3'UTR. It contained a stem-loop with a 3-way junction, a long-range and internal pseudoknot followed by a simple stem-loop element (Fig. 5A). Given the location of this structure at the position that matched the observed length of KRBV xrRNA and the organisation of this element, we assumed that this was a putative xrRNA. Although this SL had features of class 1b xrRNA, such as 3nt P1 stem and no unpaired nucleotide between P2 and P3, it lacked noncanonical A-C pairing at the base of P1 stem and had a 5nt-PK (Fig. 5A, B). This length of PK is typical for class 1a xrRNAs, while all class 1b xrRNAs characterised to date contain a very conserved 3nt PK[23]. In addition, it contained a very long P2 helix (Fig. 5A, B) similar to xrRNA1 of PCV (Fig. 4A, B), which does not occur in other members of the Flavivirus genus. The mutation that prevents the formation of 5nt long-range PK in KRBV SLI was found in in vitro XRN1 digestion assay to abolish XRN1 resistance and resulted in nearly complete degradation of XRN1-treated RNA (Fig. 5C). This indicates that the identified structural element is a key determinant of XRN1 resistance and sfRNA biogenesis in KRBV.

To determine if the identified organisation of 3'UTR and structure of xrRNA are conserved within *Anopheles* associated flaviviruses (AnFV1, AnFV2), we first performed sequence alignment (Supplementary Fig. 6A) of the entire 3'UTRs of KRBV and AnFV1 (AnFV1 and AnFV2 have identical 3'UTRs). The alignment revealed a high degree of homology at the 3'-end

of their 3'UTRs. Subsequent computational folding of the homologous region of AnFV1 demonstrated the same complex stem-loop identified in KRBV (Supplementary Fig. 6B). At the 5'-end of AnFV1 3'UTR the alignment identified the region with an almost identical sequence to KRBV xrRNA, which contained a large gap (Fig. 5D, Supplementary Fig. 6A). The in silico folding of this region revealed the structure that resembles KRBV xrRNA, including 5nt PK while having a shorter P2 helix (Fig. 5E). Based on these results, we concluded that *Anopheles*-associated ISFs have a specific 3'UTR topology, which is conserved within the group, but distinct from other flaviviruses.

In conclusion, we determined the structure of the 3'UTR of *Anopheles*-associated classical ISF. We found that it contains two conserved structural elements– one is xrRNA, and another one is TBF-like SL. The lack of xrRNA duplications in this group of viruses suggests that they are likely positioned phylogenetically prior to the evolutionary event that resulted in structure duplication and is potentially ancestral to all other ISFs. In addition, the fact that xrRNAs of *Anopheles*-associated flaviviruses share characteristics of cISF class 1b and MBF/dISF class 1a xrRNAs indicates that they may resemble the ancestral form of xrRNA, from which both subclasses of xrRNAs emerged later in the evolution.

**xrRNAs of ISFs and MBFs evolved from structures similar to xrRNAs of *Anopheles*-associated ISFs and were independently duplicated in each clade.** To examine phylogenetic relationships between XRN1-resistant elements of different ISFs, we performed

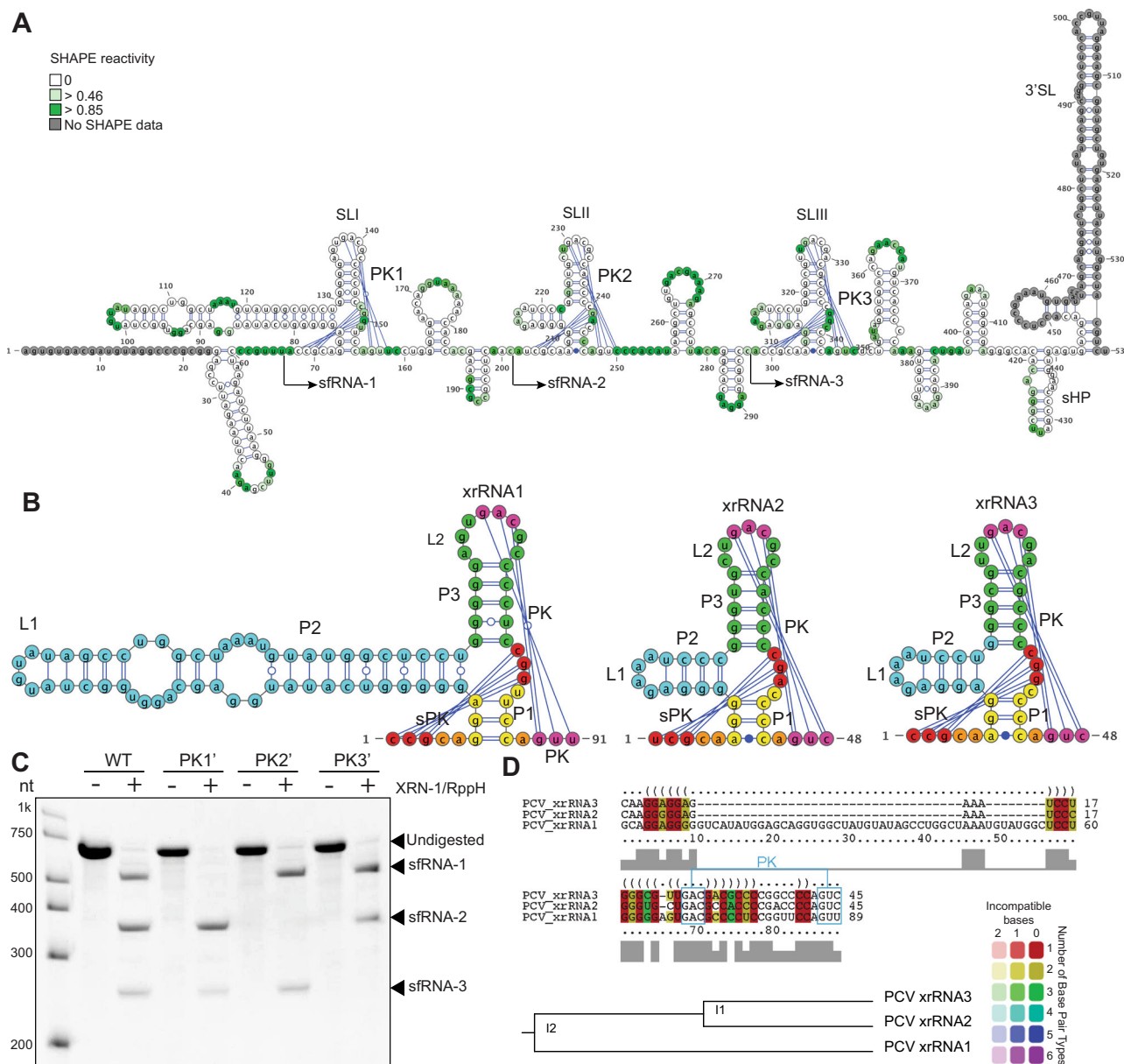

**Fig. 4 cISFs PCV contains multiple copies of divergent class 1b xrRNAs. A** Secondary structure of PCV 3'UTR generated using SHAPE-assisted folding. Shading intensity indicates NMIA reactivity. Values are the means from 3 independent experiments, each with technical triplicates. SL – stem loop, PK – pseudoknot. **B** Secondary structure of PCV xrRNAs. Each element of the secondary structure is shown in colour. **C** In vitro XRN1 resistance assay with WT and mutated PCV 3'UTRs. PK1′, PK2′ and PK3′ mutations were introduced into the terminal loop regions of SLI, SLII and SLIII, respectively, and represented the GAC - > CUG change in the PK-forming region. Image is representative from three independent experiments that showed similar results. **D** Alignment of sequence and structure of PCV xrRNAs was performed using LocARNA.

structure-based alignments of identified and predicted xrRNAs. Considering the potential origin of dISFs from MBFs, MBF xrRNAs were aligned with dISF xrRNAs, while cISF xrRNAs were aligned separately. Alignment demonstrated that dISF xrRNAs diverged from MBF xrRNAs in two independent evolutionary events (Fig. 6A). In addition, it showed that some MBF xrRNAs (e.g. DENV1 and DENV3 xrRNA-2) had higher similarity with dISF xrRNAs than with another xrRNA of the same virus (Fig. 6A). Given that dual host-associated ISFs are phylogenetically basal to the DENV group (Supplementary Figure 7), this indicates that duplication of xrRNAs in MBFs likely occurred when the DENV clade diverged. This is due to the evidence that the most basal member of the DENV group, DENV4, does not contain duplicated xrRNAs[24]. MBFs generally have a higher

similarity between corresponding xrRNAs of different viruses than between individual xrRNAs within the same virus. However, cISFs do not follow this trend and commonly exhibit higher homology between different xrRNAs of the same virus than between xrRNAs of different viruses (Fig. 6B). This indicates that duplications of xrRNAs in insect flaviviruses were more frequent, while xrRNAs of mosquito-borne viruses likely evolved after a single duplication event. In addition, structure-based alignment identified xrRNAs of *Anopheles*-associated ISFs that lack structure duplication as the most basal and likely the most ancestral to all other insect-specific flaviviruses (Fig. 6B).

To further elucidate evolutionary relationships between 3'UTR topology of insect-specific and mosquito-borne flaviviruses we conducted structure informed sequence alignment of the

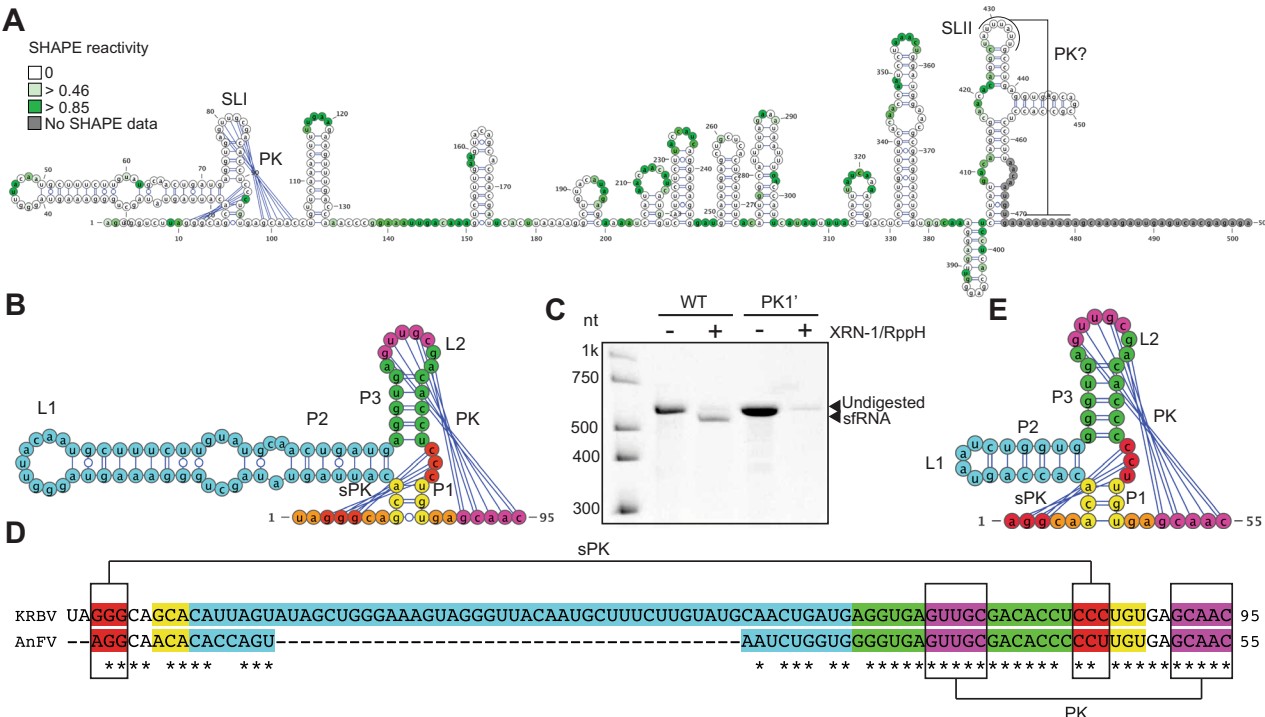

**Fig. 5** *Anopheles*-associated ISFs contain a single copy of xrRNA, which shares features of class 1a and class 1b xrRNAs. **A** Secondary structure of KRBV 3'UTR generated using SHAPE-assisted folding. Shading intensity indicates NMIA reactivity. Values are the means from 3 independent experiments, each with technical triplicates. SL – stem loop, PK -pseudoknot. **B** Secondary structure of KRBV xrRNA. Each element of the secondary structure is shown in colour. **C** In vitro XRN1 resistance assay with WT and mutated KRBV 3'UTRs. PK1' mutation was GUUGC - > CAACG change of PK-forming nucleotides in the L2 loop of the SLI. The experiment was repeated three times with similar results. **D** Sequence alignment of KRBV and AnFV1 xrRNAs. The colour coding matches (**B**) and shows conserved structural elements. **E** Predicted secondary structure of AnFV1 xrRNA. The conserved structural elements are shown in colours that match (**D**).

complete 3'UTRs of all cISFs, dISFs and MBFs and performed maximum-likelihood phylogenetic inferences. We also built covariance models for extended xrRNAs (canonical xrRNA stem-loops followed by small stem-loops) for all three clades of flaviviruses to identify their consensus structures. The 3'UTR alignment demonstrated the ancestral position of the *Anopheles* viruses to both – cISF and dISF clades (Fig. 6C, Supplementary Fig. 7). The topology of the 3'UTR phylogenetic tree was congruent with the overall topology of the viral polyprotein phylogenetic tree (Supplementary Fig. 7). This indicates the co-evolution of both the polyprotein and 3'UTR sequences. The clade of classical ISFs containing two copies of xrRNAs (CFAV, Kamiti river virus [KRV] and Aedes flavivirus [AEFV]) is more closely related to *Anopheles*-associated ISFs. In contrast, mosquito-borne flaviviruses containing multiple copies of xrRNAs have a far greater evolutionary distance (Fig. 6C, Supplementary Fig. 7). This further confirms that the evolution of classical ISFs involved two separate events of xrRNA duplications. In the MBF/dISF part of the tree, the Yellow fever group was the closest to the *Anopheles*-associated ISF-like ancestor, with both containing single xrRNA. It was followed by two branches of dISFs all containing single xrRNA and then the rest of MBFs with two xrRNA structures. Therefore, the emergence of MBF clade involved a single xrRNA duplication event. The structural alignment of individual xrRNAs (Fig. 6A and B) indicates that mosquito-borne viruses and classical ISFs did not evolve from the common ancestor with duplicated structures but rather acquired duplications due to independent evolutionary events (Fig. 6C). This also shows that dual host-associated ISFs did not lose extra copies of xrRNA while diverging from mosquito-borne viruses and transitioning from

dual-host to single-host life cycle as previously suggested. Rather, they branched off before the duplication event in the MBF clade occurred (Fig. 6C). Moreover, it appears that another duplication event occurred in the dISF clade, which resulted in two copies of xrRNAs being present in BinJV 3'UTR.

The results of phylogenetic analysis collectively indicate that classical insect-specific flaviviruses and mosquito-borne flaviviruses diverged from a common ancestor similar to *Anopheles*-associated ISFs. The evolution of insect-specific flaviviruses indicates a strong selective pressure for increasing the cumulative XRN1 resistance in the 3'UTRs either via duplication of existing xrRNAs (MBFs, cISFs) or by gaining additional pseudoknots that improve the performance of the existing XRN1-resistant structures. The fact that duplication of xrRNAs independently occurred several times in flavivirus evolution and was further selected indicates that multiple sfRNA species confer an advantage in viral fitness irrespective of which hosts they replicate.

**ISFs can tolerate the loss of individual sfRNAs, but not a complete sfRNA deficiency.** A previous study on Dengue viruses proposed that different xrRNAs within the same 3'UTR produced functionally divergent sfRNA species[10]. One of these sfRNAs (shorter) was beneficial for virus replication in the insect host, and another sfRNA (longer) for replication in vertebrates[10]. Switching between the production of longer and shorter sfRNAs was suggested as part of adaptation for virus shuttling between mosquitoes and vertebrates[10,24]. Here we demonstrated that single host (insect-specific) flaviviruses also contain duplicated xrRNAs and produce multiple sfRNA species. As ISFs don't switch hosts, the duplications of xrRNAs in these viruses should

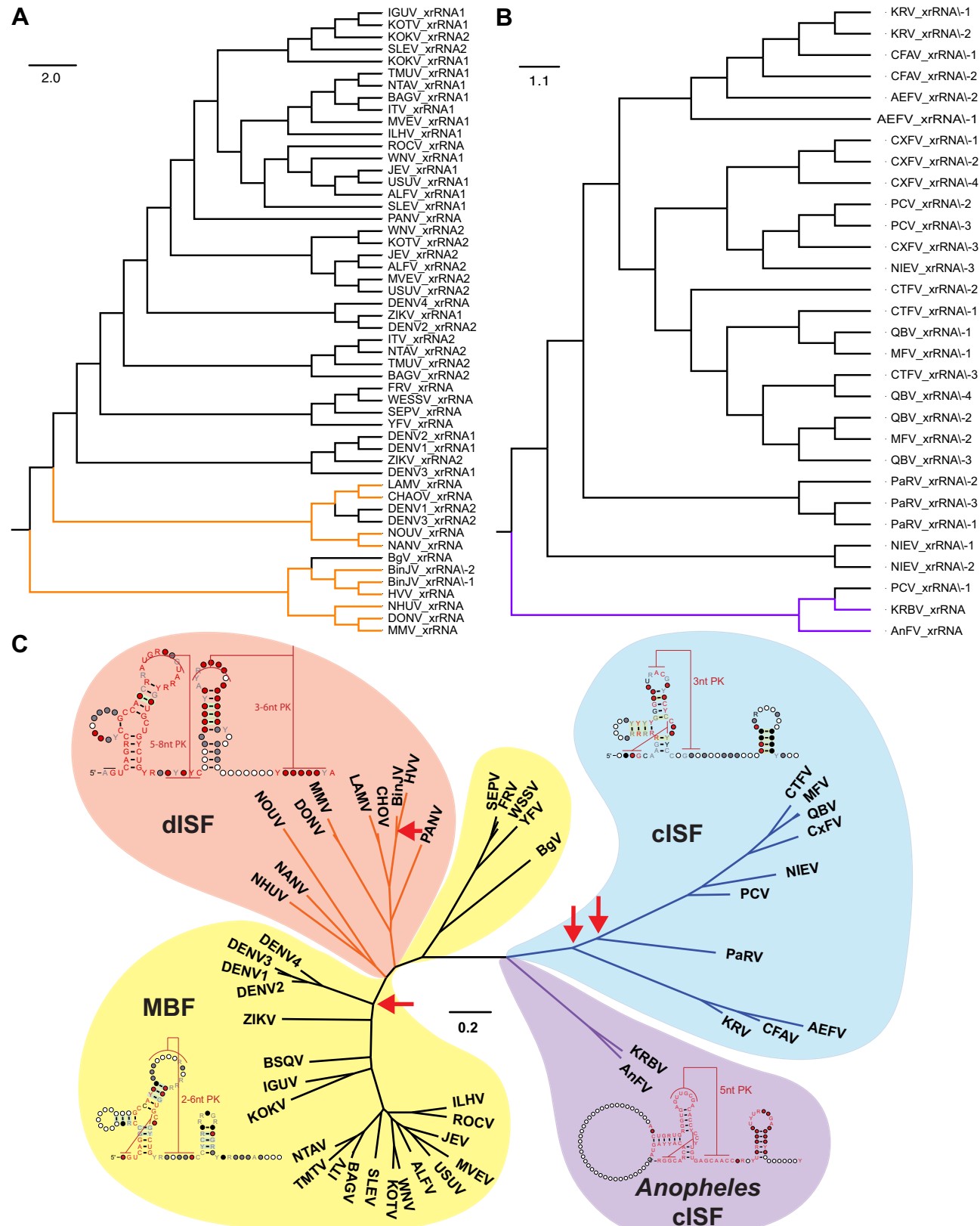

**Fig. 6 Phylogenetic analysis of flavivirus xrRNAs and 3'UTRS. A** Structure-based phylogenetic tree of cISF xrRNAs. Purple - *Anopheles*-associated ISFs. **B** Structure-based phylogenetic tree of MBF and dISF xrRNAs. Orange - dISF clades. **C** Consensus Maximum-Likelihood phylogenetic tree of flavivirus 3'UTRs. Arrows indicate xrRNA duplication events.

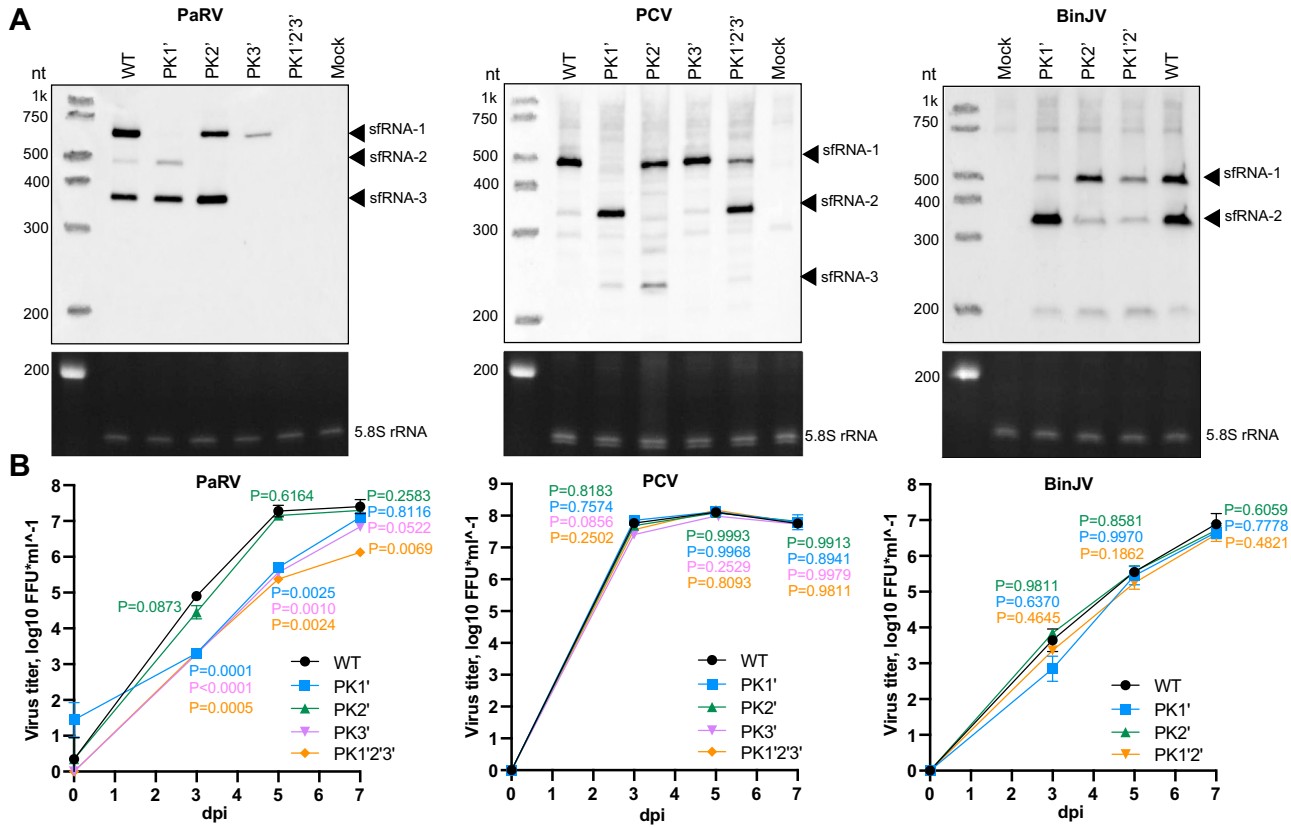

**Fig. 7 Effect of sfRNA deficiency on replication of ISFs in mosquito cells. A** Northern blot analysis of sfRNA production by WT and PK mutants of PARV, PCV and BinJV. C6/36 cells were infected with WT or mutant viruses at MOI = 1. Total RNA was isolated at 7 dpi and used for blotting. The bottom panel show Et-Br-stained 5.8 S rRNA as an equal loading control. Images are representative of two independent experiments that produced similar results. PK1', PK2, and PK3' mutations represented substitutions of PK-forming nucleotides in L2 loops of SLI, SLII and SLII, respectively, with complementary nucleotides and are the same as in Fig. 2D (BinJV), 3 C (PaRV), 3 G (PCV). **B** Virus titers in culture fluids of RML-12 cells infected with WT or PK-mutants of ISFs. Cells were infected at MOI = 0.1, culture fluids were sampled at the indicated time point, and titers were determined using foci-forming immunoassay on C6/36 cells. Values are the means from three biological replicates ± SD. Statistical analysis was performed by two-way ANOVA with Dunnett's correction, all comparisons were to WT, test was two-sided.

have a different biological function. It is also unclear whether different sfRNA species produced by ISFs are functionally redundant or divergent.

To clarify this matter, we generated mutant ISFs deficient in the production of sfRNAs and assessed their replication. If sfRNA isoforms are redundant, we expected that viruses would tolerate deficiency in single sfRNAs but become attenuated when production of all sfRNAs is impaired. If individual sfRNA species have unique, specialised functions, we expected that loss of each sfRNA should impair virus replication. Therefore, we introduced mutations that prevent PK formation by SLI, SLII and SLIII of PaRV and PCV generating mutants in PK1', PK2' and PK3', respectively. We also generated combined PK1'2'3' mutants for PaRV and PCV in which all three PKs were mutated. As BinJV contains only two canonical xrRNAs, we generated PK1', PK2' and PK1'2' mutants for this virus. The mutations were the same as those analysed in in vitro XRN1 resistance assays.

Northern blot analysis of RNA isolated at 7dpi demonstrated that PK1', PK2' and PK3' mutations abolished production of sfRNA-1, sfRNA-2 and sfRNA-3, respectively, by PaRV and PCV (Fig. 7A). For PaRV, PK3' mutation also reduced sfRNA1 and sfRNA2, suggesting some long-range interactions between xrRNAs. As expected, triple mutant PK1'2'3' of PARV was completely deficient in the production of all three sfRNAs (Fig. 7A). However, two attempts to recover PK1'2'3' mutant of

PCV yielded very low viral titers (~$10^3$ FFU/mL vs ~$10^{6-7}$ FFU/mL observed for other mutants and WT virus). Moreover, sequencing of the recovered virus revealed reversion of mutations in xrRNA1 and xrRNA2 that restored the production of sfRNA-1 and sfRNA-2, respectively (Fig. 7A). This indicates that complete sfRNA deficiency is detrimental for PCV replication and reveals a strong selective pressure to restore sfRNA production. In the case of BinJV, PK1' and PK2' mutants had decreased but not completely abolished production of corresponding sfRNAs, consistent with the in vitro assay results (Fig. 2D) and explained by the additional XRN1 resistance conferred by the novel pseudoknots (Fig. 2D, E). Accordingly, the PK1'2' mutant of BinJV showed reduced but not completely abolished production of both sfRNA-1 and sfRNA-2.

To determine how complete and partial deficiency in sfRNA affects replication of ISFs, we compared growth kinetics of generated mutants and corresponding WT viruses in RNAi-competent mosquito cells RML-12. For PaRV, for which we obtained a complete set of sfRNA-deficient phenotypes, the lack of sfRNA-2 (PK2' mutant) did not affect virus replication (Fig. 7B). However, this can be attributed to the overall low abundance of this sfRNA even in the WT virus which is caused by the unusual fold of xrRNA2 (Figs. 1A, 1C, 7A). PaRV mutants deficient in sfRNA-1 (PK1' mutant), sfRNA1 and sfRNA-3 (PK3' mutant – no sfRNA3 and much less sfRNA1), and all three

sfRNAs (PK1'2'3' mutant) were attenuated at 3dpi and 5dpi (Fig. 7B). However, by 7dpi, PK1' and PK3' mutants overcame the initial attenuation and reached replication levels comparable to the WT virus, while replication of PK1'2'3' mutant remained significantly impaired (Fig. 7B). This indicates that the loss of individual sfRNAs could be tolerated. In contrast, the loss of all three sfRNAs substantially reduced virus replication through the entire course of infection. Notably, individual PK mutants of PCV replicated at the same levels as WT virus (Fig. 7B), indicating that deficiency in individual sfRNAs is tolerable for ISFs and suggests that their functions are likely to be redundant. Moreover, the recovery of sfRNA1 and sfRNA2 production by PK1'2'3' mutant of PCV due to reversions restored replication of this virus. PK1', PK2' and PK1'2' mutants of BinJV also replicated at the levels comparable to the WT virus at all time points (Fig. 7B). However, these mutant viruses were not fully deficient in either sfRNA-1 or sfRNA-2 (Fig. 7A), consistent with the in vitro XRN1 digestion assay (Fig. 2D, E). This further demonstrates the functional significance of novel pseudoknots for enabling the production of sfRNAs (albeit at reduced levels) when xrRNA1 and xrRNA2 are affected by mutations (Fig. 7A).

Collectively, our results indicate that sfRNA species of different lengths are likely to be functionally redundant in ISFs. Therefore, we propose that duplication of xrRNAs ensures sfRNA production even if one of the structures loses XRN1 resistance. We also showed that in dISF, a similar backup mechanism for sfRNA production is provided by novel pseudoknots located downstream of the canonical xrRNAs. The presence of these novel pseudoknots, which is unique for dISFs, explains why most dISFs tolerated the loss of duplicated xrRNAs without compromising the robustness of sfRNA production and virus viability.

## Discussion

All dual host flaviviruses tested to date were shown to produce sfRNAs by employing XRN1-resistant structured RNA elements[4]. Herein we demonstrated that classical and dual host-associated insect-specific flaviviruses also produce sfRNAs by employing the XRN1-resistance mechanism. Using RNA SHAPE we, for the first time, determined the secondary structures of complete 3'UTRs in the divergent representatives of cISF and dISF lineages and experimentally identified structured RNA elements that enable XRN1 resistance and production of sfRNAs in these viruses. Conservation of sfRNA production in the highly divergent classical ISFs, dual host-associated ISF BinJV and Anopheles-associated cISF KRBV indicates the crucial importance of sfRNAs for flavivirus replication in insect hosts. Moreover, considering that Anopheles-associated ISFs are believed to be the most closely related viruses to the common flaviviral ancestor[33], the production of sfRNA by KRBV indicates that the ability to resist RNA degradation by XRN1 and generate sfRNAs was fixed very early in flavivirus evolution. This further emphasises the importance of sfRNAs in the life cycle of flaviviruses.

Most mosquito-borne vertebrate infecting flaviviruses are known to produce at least two sfRNA species due to the presence of duplicated xrRNAs in their 3'UTRs[4], while ISFs were predicted to lack these duplications[24,28]. In addition, a previous study demonstrated that adaptation of DENV to insect and vertebrate hosts involves switching between the production of shorter and longer sfRNA species, respectively, due to the accumulation of point mutations in corresponding xrRNAs[10,37]. Longer sfRNA species benefited DENV replication in vertebrates, while shorter sfRNAs were shown to function in mosquito cells[10]. It was, therefore, suggested that duplication of xrRNAs occurred in dual host infecting flaviviruses to enable switching between the production of functionally distinct sfRNAs while adapting to

replication in different hosts. Based on these observations, the lack of xrRNA duplications in ISFs was proposed as a reason for their host restriction.

Contrary to this, we found that all ISFs analysed in our study except KRBV contained duplicated xrRNAs and produced multiple sfRNAs, although none of the viruses could infect vertebrate cells. Moreover, representatives of the YFV-group[20] and the DENV4[38] virus contain only one xrRNA while replicating in both insect and vertebrate hosts. Collectively, published data and our results herein indicate that duplication of xrRNAs has not resulted from adaptation to the dual host life cycle and that lack of these duplications is not the mechanism of host restriction in ISFs. Furthermore, we found that deficiency in the production of individual sfRNAs can be tolerated by ISFs, while the deficiency in all sfRNAs is largely detrimental to ISF replication fitness. These results are consistent with previous observations in MBFs. In particular, Zika virus and West Nile virus mutants deficient in single sfRNAs replicated in mosquito and vertebrate cells at the levels comparable to WT virus, whereas mutations that abolished production of both sfRNAs either resulted in a substantial attenuation of virus replication or were lethal[7,8,15,39,40]. Therefore, we concluded that in many flaviviruses, including ISFs, different sfRNA species are functionally redundant and that host specialisation of sfRNAs observed for DENV1-3 viruses[10,37] is likely unique to these members of DENV serogroup.

Another important novel outcome of our study is the discovery of the ancestral form of xrRNA in the 3'UTR of Anopheles-associated cISF KRBV. Together with the structural data for the 3'UTR of KRBV and other ISFs generated here, it enabled us to reconstruct the most complete to date structure-based phylogenetic trees of flavivirus xrRNAs and 3'UTRs and mapping of xrRNA duplication events on them. The Anopheles-associated ISFs contain a single copy of the XRN1-resistant element, which has a general topology of class 1b xrRNA (present in TABV and cISFs) while lacking noncanonical C-A pairing and forming 5-nt pseudoknot, which only occurs in class 1a xrRNAs of MBFs and dISFs. This structure was found to be ancestral for both cISFs and MBFs/dISFs xrRNAs and likely has the same organisation as xrRNA of a common flavivirus ancestor. Notably, viruses from YFV serogroup were most closely related to Anopheles-associated viruses on the MBF branch of the phylogenetic tree (Fig. 6C). They were followed by two independent branches of dISFs and then the rest of MBFs. As viruses from YFV serogroup do not contain xrRNA duplications, this indicates that dISFs diverged before the duplication event in the MBF clade and an independent xrRNA structure duplication occurred later in evolution in the BinJV. On the cISF branch, all viruses were found to contain duplicated xrRNAs, which indicates that structure duplications occurred very early in the evolution of ISFs. Based on this analysis, we concluded that duplicated xrRNAs in the 3'UTRs of phylogenetically distant flaviviruses were not the result of divergence from a common ancestor. Instead, they appeared due to convergent evolution with independent duplication events in cISF, MBF and dISF clades. Furthermore, the viruses that contain duplications were strongly selected for in both cISF and MBF lineages, indicating that the presence of multiple xrRNAs has a strong adaptive advantage for both insect-specific and dual host virus life cycles.

Given the functional redundancy of sfRNA species produced from duplicated xrRNAs in ISFs, we propose two potential mechanisms of how this redundancy could benefit viral replication. The robust production of sfRNA is crucial for the efficient replication of flaviviruses. However, individual xrRNAs are dynamic structures that can transition between XRN1-resistant to XRN1 sensitive conformation[41]. Therefore, the most apparent benefit from xrRNA duplication is that downstream copies of

xrRNA ensure that sfRNA is produced when upstream xrRNA is not folded into its XRN1-resistant state. The fact that shorter sfRNAs are always observed in infected cells along with longer sfRNAs indicates that the most upstream longer xrRNA does not often confer complete XRN1 resistance, thus allowing XRN1 to progress to the subsequent xrRNA, generating a shorter sfRNA variant. Hence, duplications of xrRNAs enable fail-safe sfRNA production despite the thermodynamic fluctuations in individual xrRNAs. Another evolutionary benefit from duplicated xrRNAs is protection from mutations in individual XRN1-resistant elements. This mechanism is supported by structural features identified in the 3′UTR of PaRV. This UTR contains four copies of pseudoknot-forming stem-loops (SL-PK) that have homologous sequence and structure. However, only two (SLI-PK1 and SLIII-PK3) confer efficient XRN1 resistance and generate substantial amounts of sfRNAs. At the same time, SLIV-PK4 has lost its XRN1 resistance completely, while SLII-PK2 accumulated mutations that rendered it significantly less resistant to XRN1 digestion. Despite these mutations, PaRV retained its viability due to the production of sfRNAs from the other two copies of xrRNAs, xrRNA1 and xrRNA3. Based on these data, we concluded that xrRNA duplications improve viral fitness by providing a backup strategy for sfRNA production.

Despite the clear benefit of more robust sfRNA production due to the structure duplications in the 3′UTR, there are two groups of flaviviruses (YFV serogroup and the majority of dISFs) that are evolutionary successful while lacking additional copies of canonical xrRNAs. Our results indicate that dISFs evolved a different strategy to backup sfRNA production. While all dISFs, except BinJV, don't have additional copies of canonical xrRNA, we found that they all contain pseudoknot(s) formed by a conserved stem-loop elements (CS3/RCS3 in BinJv), which is not evident in other flavivirus clades. Our mutational analysis in the live BinJV demonstrated that the presence of the additional pseudoknots ensures the production of sfRNAs in amounts sufficient to prevent virus attenuation when the canonical pseudoknots of the corresponding xrRNAs are mutated. Curiously, novel pseudoknots of dISFS are not XRN1-resistant on their own and likely act by stabilising the resistant conformation of the upstream RNA elements. However, further structural studies are required to provide mechanistic insights on how these novel pseudoknots affect the folding of dISF xrRNAs. In particular, solving the crystal structure of dISF xrRNAs with and without nPKs will be desirable in the future to provide such information.

The discovery of novel pseudoknots in dISFs and independent xrRNA duplication events in all flavivirus clades shows that flaviviruses evolved multiple strategies to ensure the reliable production of sfRNAs. Together with highly attenuated phenotypes of viruses completely deficient in sfRNA production, this further highlights the critical importance of sfRNAs for replication of all flaviviruses, including ISFs. Identifying the exact functions of sfRNAs in ISF-host interactions is the next step to understand their functional significance. This can be achieved by further exploring the biological effects of ISF sfRNAs in the loss of function model systems employing ISF mutants deficient in sfRNA production. We and others previously employed such systems to identify molecular functions of MBF sfRNAs[7–9,15] and, through this study, are now also available for ISFs.

## Methods

**Cell culture**. *Aedes albopictus* larvae cells C6/36 (ATCC – CRL-1660) and *Aedes aegypti* larvae cells Aag2 (ATCC – CCL-125) were obtained from the ATCC. *Aedes albopictus* larvae cells RML-12[42] were a gift from Prof. Robert Tesh (UTMB, USA). C6/36 were cultured in Roswell Park Memorial Institute 1640 medium (RPMI 1640). Aag2 cells were cultured in 1:1 mixture of Schneider's *Drosophila* medium and Mitsuhashi & Maramorosch medium (Sigma, USA). RML-12 cells were cultured in Leibovitz's L-15 medium supplemented with 10% tryptose-phosphate

broth. All culture media were supplemented with 10% Fetal Calf Serum (FCS), 100 μg/mL streptomycin, 100 U/mL penicillin, and 2 mM L-glutamine. Cells cultured at 28 °C with 5% $CO_2$ or in sealed containers. All cell culture media and reagents were from Gibco, USA, unless otherwise specified.

**Processing of mosquitoes**. All of the mosquito samples tested in this study were archival and collected from a previous study[33]. Mosquitoes were snap-frozen in liquid nitrogen and shipped for processing on dry ice. Total RNA was then isolated from individual mosquitoes and used for RT-PCR screening with Karumba virus (KRBV) genome primers KRBV-F and KRBV-R (Supplementary Table 2). RT-PCR was performed using SSIII One-Step RT-PCR Kit with Platinum Taq (Invitrogen, USA) according to the manufacturer's recommendations. Each RT-PCR reaction contained 1 μg of mosquito RNA. The cycling conditions were 15 min at 60 °C, 2 min at 95 °C; 40 cycles of 15 s at 95 °C, 30 s at 51 °C, 35 s at 68 °C; and a final extension for 5 min at 68 °C. PCR products were then separated in 2% agarose gel, and DNA was extracted from the gel using MinElute Gel Extraction Kit (Qiagen, Germany). Purified DNA was Sanger sequenced and matched to KRBV RefSeq sequence NC_035118.1. RNA samples confirmed to contain KRBV were used for further analyses.

**Viruses and infection**. Binjari (BinJV), Parramatta River (PaRV), Palm Creek (PCV) and Hidden Valley (HVV) viruses were previously isolated from Australian mosquito populations[29–32] and passaged in C6/C6 cells for a total of 3–6 passages. For the generation of the virus stocks used in the study C6/36 cells were infected at MOI = 0.1, and culture fluids were harvested at 7dpi. Culture fluids were then cleared from cell debris by centrifugation at 3000 x g for 15 min at 4 °C and stored at −80 °C in single-use aliquots. Virus titers were determined by foci-forming immunoassay on C6/36 cells. All infections were performed at the indicated Multiplicity of Infection (MOI) by incubating cells with 50 μL of inoculum per cm² of growth area for 1 h at 28 °C. Inoculated cells were then maintained in the growth medium containing a reduced amount of FCS (2%) to prevent overgrowth.

**Foci-forming immunoassay**. Ten-fold serial dilutions of culture fluids were prepared in RPMI media supplemented with 2% FCS, and 25 μl of each dilution were used to infect $10^5$ C6/36 cells grown in 96-well plates. After 2 h incubation with the inoculum, 180 μl overlay media was added to the cells and incubated at 27 °C in 5% $CO_2$. The overlay media contained one part X M199 medium (containing 5% FCS, 100 μg/mL streptomycin, 100 U/ml penicillin, and 2.2 g/L $NaHCO_3$) and another part 2% carboxymethyl cellulose (Sigma-Aldrich, USA). At 3 days post-infection, cells were fixed with 100 μL/well of 80% acetone for 20 min at −20 °C, washed with PBS, thoroughly dried and blocked for 30 min with 150 μL/well of ClearMilk blocking solution (Pierce, USA). Cells were then incubated with 50 μL/well of mouse monoclonal antibody to flavivirus envelope protein (4G2 for detection of BinJV[31] and HVV[32], 7D11 for PaRV[43] and 5G12 for PCV[43]), diluted in 1:100 for 1 h, followed by 1 h incubation with 50 μl/well of 1:800 dilution of goat anti-mouse IRDye 800CW secondary antibody (LI-COR, USA). All antibodies were diluted with Clear Milk blocking buffer (Pierce, USA), and incubations were performed at 37 °C for 1 h. After each incubation with antibody, plates were washed 5 times with phosphate buffered saline (PBS) containing 0.05% Tween 20 (PBST). Plates were then scanned using an Odyssey CLx Imaging System (LI-COR) (42 μm; medium; 3.0 mm). Virus replication foci were counted using the Image Studio Lite software (v5.2.5, LI-COR, USA), and titres were determined based on dilution factors and expressed as focus forming units per mL (FFU mL^-1).

**RNA isolation**. Total RNA was isolated from cells and mosquitoes using TRIreagent (Sigma, USA). Individual mosquitoes were homogenised in 500 μL TRIreagent for 5 min at 30 Hz using a Tissue Lyser II (Qiagen, USA) before RNA isolation. Viral RNA from cell culture fluids was isolated using QIAamp Viral RNA Mini Kit (Qiagen, USA). All RNA isolation procedures were conducted according to the manufacturer's instructions. RNA concentrations were determined on NanoDrop One microspectrophotometer (ThermoFisher Scientific, USA), and RNA purity was assessed by $OD_{260}/OD_{280}$ and $OD_{260}/OD_{230}$ ratios.

**Northern blotting**. Total RNA (2.5-10 μg) was mixed with an equal volume of Loading Buffer II (Ambion, USA), denatured at 85 °C for 5 min and chilled on ice for 2 min. Samples were then subjected to electrophoresis in 6% polyacrylamide TBE-Urea gels (Invitrogen, USA). Electrophoresis was performed for 90 min in 1x Tris-Borate-EDTA buffer pH8.0 (TBE). Gels were stained with ethidium bromide to visualise rRNA and documented using Omnidoc imager (Cleaver Scientific, UK). RNA was then electroblotted onto Amersham Hybond-N⁺ nylon membrane (GE Healthcare, USA) for 90 min at 35 V in 0.5x TBE using the TransBlot Mini transfer apparatus (Bio-Rad, USA) and UV-crosslinking at 1200 kDj/cm². Membranes were pre-hybridised at 50 °C in ExpressHyb Hybridization Solution (Clontech, USA) for 1 h. The probes were prepared by end labelling 10 pmoles of DNA oligonucleotide complementary to the sfRNA (Supplementary Table 2) with [γ-$^{32}$P]-ATP (Perkin-Elmer, USA) using T4 polynucleotide kinase (NEB, USA) and purified from unincorporated nucleotides by gel filtration on Illustra MicroSpin G-25 Columns (GE Healthcare, USA). Hybridisation was performed overnight at 50 °C in ExpressHyb Hybridisation Solution (Clontech, USA). After hybridisation,

membranes were rinsed, washed 4 × 15 min with Northern Wash Buffer (1% sodium dodecyl sulphate [SDS], 1% saline-sodium citrate [SSC]) at 50 °C and exposed to a phosphor screen (GE Healthcare, USA) overnight. Signal detection was performed on Typhoon FLA 7000 Imager (GE Healthcare, USA).

**RNAi knock-down.** The dsRNA against *Ae. aegypti* XRN1/**pacman** was generated by in vitro transcription (IVT). The DNA templates for IVT were prepared by amplifying a ~3 kb fragment of XRN1/**pacman** mRNA with primers designed to incorporate T7 promoter into PCR-products in either forward or reverse directions (Supplementary Table 2) to generate templates for the synthesis of sense and antisense RNA strands. Total RNA (1 μg) isolated from Aag2 cells was used as a template for RT-PCR performed using SSIII One-Step RT-PCR Kit with Platinum Taq (Invitrogen, USA) according to the manufacturer's recommendations. The cycling conditions were 15 min at 60 °C, 5 min at 94 °C; 5 cycles of 15 s at 95 °C, 30 s at 50 °C, 3 min at 68 °C; 35 cycles of 15 s at 95 °C, 30 s at 60 °C, 3 min at 68 °C and a final extension for 5 min at 68 °C. PCR products were then separated in 1% agarose gel, and DNA was extracted from the gel using Monarch Gel Extraction Kit (NEB, USA). In vitro transcription was then performed using 1 μg of purified DNA templates and MEGAscript T7 Transcription Kit (Invitrogen, USA). RNA was purified by LiCl precipitation, 50 μg of each ssRNA were combined in a total volume of 100 μL and denatured by heating at 85 °C for 10 min followed by gradual cooling to RT to produce dsRNA. The RNA annealing was validated by electrophoresis in 1% native agarose gel. Aag2 cells were then transfected with 2 μg of the resulting dsRNA per 10⁶ cells using Lipofectamine 2000 (Invitrogen, USA) following transfection in suspension protocol[44]. Cells were plated into 6-well plates and at 24 h post-transfection infected with PaRV, PCV or BinJV at MOI = 1. At 2 dpi cells were lysed in TRIreagent (Sigma, USA), and total RNA was isolated.

**3′UTR cloning.** Total RNA was isolated from C6/36 cells infected with PaRV, PCV, BinJV or HVV at 5dpi and from KRBV-positive mosquitoes, and RT-PCR was used to amplify the fragment of the viral genome containing 3′UTR and the small terminal part of the viral NS5 gene. RT-PCR was performed using SSIII One-Step RT-PCR Kit with Platinum Taq (Invitrogen, USA) according to the manufacturer's recommendations. PCR primers (Supplementary Table 2) were 5′-phosphorylated, and forward primers were designed to incorporate T7 promoter at the 5′-end of the amplicons. The cycling conditions were 15 min at 60 °C, 2 min at 95 °C; 35 cycles of 15 s at 95 °C, 30 s at 55 °C, 40 sec at 68 °C; and a final extension for 5 min at 68 °C. PCR products were then separated in 2% agarose gel, purified using Monarch Gel Extraction Kit (NEB, USA) and ligated with SmaI-digested and dephosphorylated pUC19 vector using Blunt/TA Ligase Mastermix (NEB, USA). Ligation was performed overnight at 16 °C, and products were transformed into NEB5α chemically competent cells (NEB, USA) and plasmids were isolated using QIAprep Spin Miniprep Kit (Qiagen, Germany).

**XRN1 resistance assay.** Plasmids were linearised by restriction digest and purified using Monarch PCR and DNA Clean-up Kit (NEB, USA). 3′UTRs were in vitro transcribed from 1 μg of linearised plasmids using MEGAscript T7 Transcription Kit (Invitrogen, USA) according to the manufacturer's recommendations. RNA was purified by LiCl precipitation and analysed by electrophoresis in a 1.2% denaturing agarose gel. RNA was then refolded in NEB3 buffer by heating at 85 °C for 5 min followed by gradual cooling to 28 °C. The refolded RNA (1 μg) was incubated with 1U XRN1 (NEB, USA) and 10U RppH (NEB, USA) in 20 μL of reaction mixture containing 1x NEB3 buffer (NEB, USA) and 1 u/μL RNasin RNase Inhibitor (Promega, USA). Incubation was performed for 2 h at 28 °C. The reaction was stopped by adding 20 μL of Loading Buffer II (Ambion, USA), heating for 5 min at 85 °C and placing on ice. The entire volume was then loaded into 6% polyacrylamide TBE-Urea gels (Invitrogen, USA), and electrophoresis was performed for 90 min in 1xTBE. Gels were stained with ethidium bromide and documented using an Omnidoc imager (Cleaver Scientific, UK).

**RNA ligation-mediated RT-PCR method for sequencing of sfRNA ends.** Total RNA from infected C6/36 cells (20 μg) was denatured by heating at 75 °C for 5 min, then placed on ice. Denatured RNA was circularised by incubation with 10U T4 RNA Ligase 1 (NEB, USA) in 20 μL of the reaction mixture containing 1x T4 RNA Ligase Buffer (NEB, USA), 1 mM ATP (NEB, USA), 12.5% PEG8000 and 1 u/μL RNasin RNase Inhibitor (Promega, USA). Incubation was performed O/N at 16 °C, then 10 μL of the mix was used as the template for RT reaction with LigSeq-RT primer (Supplementary Table 2) and SuperScript IV RT enzyme (Invitrogen, USA), which was performed according to the manufacturer's recommendations. RNA was then removed from the mixture by incubation with 5U RNase H (NEB, USA) and 10 μg DNase-free RNase A (ThermoFisher Scientific, USA) for 20 min at 37 °C. RNA-free cDNA (5 μL) was used as a template for PCR with PrimeStar GXL Polymerase (Takara, Japan) and back-to-back PCR primers (LigSeq-F and LigSeq-R, Supplementary Table 2) designed in the proximity to the 3′-end of viral 3′UTRs. The cycling conditions were 1 min at 98 °C; 40 cycles of 15 s at 98 °C, 45 sec at 68 °C; and a final extension for 5 min at 68 °C. Amplicons were separated in 2% agarose gel, purified using Monarch Gel Extraction Kit, and Sanger sequenced with LigSeq-F and -R primers. The resulted sequences were aligned to the reference genomes using CLC Main Workbench v8.1.0 (Qiagen, Germany) and nucleotide

positions found to be ligated to the known 3′-end of the 3′UTRs were identified as 5′-ends of the sfRNAs.

**Selective 2′ Hydroxyl Acylation analyzed by primer extension (SHAPE).** In vitro transcribed 3′UTR RNA (20 μg) dissolved in 20 μL DEPC-H₂O was heated at 95 °C for 2 min, then placed on ice for 2 min. RNA was then refolded by incubation at 28 °C for 30 min in 150 μL of RNA folding buffer (100 mM HEPES pH 8.0, 100 mM NaCl, 6 mM MgCl₂, 0.5 U/μL RNasin PLUS RNase Inhibitor (Promega, USA)). The solution was equally divided between two tubes (72 μL each) and incubated for 30 min at 28 °C with either 8 μL of 50 mM NMIA (ThermoFisher Scientific, USA) in DMSO ( + reaction) or 8 μL of DMSO (- reaction). RNA solutions were mixed with 4 μL 3 M NaOAc pH 5.2, 2 μL 100 mM EDTA, 1 μL Glycogen (20 mg/mL) and 350 μL ice-cold 100% ethanol and incubated overnight at −80 °C to precipitate RNA. RNA was pelleted by 30 min of centrifugation at 10,000 x g, 4 °C, washed with 75% DEPC-EtOH and resuspended in 10 μL 0.5x TE buffer pH 8.0. Five microliters of each RNA solution were combined with 6 μL DEPC-H₂O and 1 μL of 5′-FAM-labelled SHAPE primer (Supplementary Table 2). Primers were annealed to the template by incubation for 1 min at 85 °C, 10 min at 60 °C and 10 min at 35 °C. Each reaction was then combined with 1 μL DTT (Invitrogen, USA), 0.5 μL RNasin PLUS RNase Inhibitor (Promega, USA), 0.5 μL DEPC-H₂O, 1 μL 10 mM dNTPs, 1 μL SuperScript III RT (Invitrogen, USA) and 4 μL SSIII First Strand Buffer (Invitrogen, USA). RT reaction was performed at 52 °C for 30 min and quenched by adding 1 μL 4 M NaOH followed by incubation at 95 °C for 3 min. NaOH was neutralised by adding 2 μL of 2 M HCl. Reactions were subsequently chilled on ice and EtOH-precipitated overnight at −80 °C as described above. Pelleted cDNA was dissolved in 40 μL of deionised formamide at 65 °C for 10 min and then stored at −80 °C.

To prepare a sequencing ladder, 10 μg RNA was incubated with 10 pmol 5′-HEX-labelled SHAPE primer (Supplementary Table 2) in 10 μL volume a for 1 min at 85 °C, 10 min at 60 °C and 10 min at 35 °C. Each reaction was then combined with 1 μL DTT (Invitrogen, USA), 0.5 μL RNasin PLUS RNase Inhibitor (Promega, USA), 0.5 μL DEPC-H₂O, 1 μL 10 mM dNTPs, 1 μL of 10 mM ddTTP (Roche, USA), 1 μL SuperScript III RT (Invitrogen, USA) and 4 μL SSIII First Strand Buffer (Invitrogen, USA). The cDNA was then synthesised and precipitated as described above for the SHAPE reactions. Pelleted cDNA was dissolved in 40 μL of deionised formamide at 65 °C for 10 min, and 10 μL of the solution was combined with 40 μL of (+) and (-) SHAPE cDNAs. Samples were then analysed by high throughput capillary electrophoresis. Chromatogram data was analysed, and SHAPE reactivities were computed using QuShape software[45]. Mean SHAPE reactivities from three independent experiments were used as constraints for guided RNA folding with RNAStructure software (https://rna.urmc.rochester.edu/RNAstructureHelp.html). The folding temperature was set to 28 C, folding of pseudoknots was enabled, and the maximum distance of base pairing was set to 100 nt. Resultant secondary structures were visualised using VARNA v3.93. Noncanonical C-A pairing in cISF sfRNAs was forced manually.

**RNA structure prediction.** Pseudoknots if dISF xrRNAs were predicted using IPknot[46] web server (http://rtips.dna.bio.keio.ac.jp/ipknot/). Secondary structures of AnFV 3′UTR elements were predicted using *mfold* v2.3 (http://www.unafold.org/mfold/applications/rna-folding-form-v2.php), and pseudoknots were located manually. The folding temperature was set to 28 °C, and the maximum distance between paired nucleotides was limited to 150. All secondary structures were visualised using VARNA v3.93.

**Multiple sequence alignment, covariance analysis, and phylogenetic inference.** Structure-based multiple alignments of xrRNAs were performed using RIBOSUM-like similarity scoring implemented in the LocARNA package (Freiburg RNA Tools)[47]. Experimentally determined or predicted secondary structures were provided for each xrRNAs as structural constrains (#S option). Stockholm formatted alignments and consensus secondary structures generated by LocARNA were used to build covariance models followed by evaluation and visualisation of consensus structures with R-scape software[36] (http://eddylab.org/R-scape/). Consensus pseudoknots were specified manually in R-scape input as consensus structures #CS_2 and #CS_3.

Multiple sequence alignment (MSA) used for phylogenetic inference were constructed using iterative refinement methods incorporated into MAFFT v7.475[48] for both the polyprotein (G-INS-i) and complete 3′UTR (Q-INS-i; which incorporates structural information[49] of 48 representative Flaviviruses. The resultant MSA dimensions were (3897 × 48) and (1655 × 48) for the polyprotein and nucleotide, respectively, and accession IDs are available from Supplementary Table 1. IQ-TREE2 (v2.1.2) ModelFinder, tree search, SH-aLRT test and ultrafast bootstrap[50] were used to construct the consensus maximum-likelihood phylogenetic inference (Command: --alrt 1000 -B 1000). The best-fit protein substitution model (LG + F + R6) and nucleotide substitution model (GTR + F + I + G4) was selected using the ModelFinder program within IQ-TREE2 informed using the Bayesian Information Criterion. The resultant consensus tree was visualised using FigTree v1.4.4 (Rambaut, A., 2021, https://github.com/rambaut/figtree/).

**PCR-directed mutagenesis and generation of mutant viruses**. Plasmid constructs based on pUC19 vector containing WT 3'UTR of each ISF were used as templates for PCR-directed site-specific mutagenesis, which was performed using Q5 Site-Directed Mutagenesis Kit (NEB, USA) with mutagenesis primers listed in Supplementary Table 2. Cycling conditions for mutagenesis PCR were were 1 min at 98 °C; 25 cycles of 10 s at 98 °C, 30 s at 55 °C, 2 min at 72 °C; and a final extension for 2 min at 72 °C. PCR products were analysed by electrophoresis in 1% agarose gel, and if multiple amplification products were evident, individual amplicons were gel-purified using QIAquick Spin Gel extraction kit (Qiagen, Germany). Products of PCR amplification were subjected to KLD reaction using Q5 Site-Directed Mutagenesis Kit (NEB, USA) and transformed into NEB 5-alpha Competent E. coli High Efficiency (NEB, USA) according to the manufacturer's instructions. Plasmids were isolated from 5 mL of overnight liquid culture using QIAprep Spin Miniprep Kit (Qiagen, Germany), and Sanger Sequencing confirmed the presence of mutations.

Mutant viruses were assembled using circular polymerase extension reaction (CPER) according to the established protocols for PaRV, PCV and BinJV. CPER fragments containing WT or mutated or 3'UTRs were amplified from the plasmids using PrimeStar GXL polymerase (Takara, Japan) and gel-purified. Infectious cDNA from CPER assembly was transfected into C6/36 cells using Lipofectamine 2000 (Invitrogen, USA) according to the manufacturer's instructions. Culture fluids of transfected cells were then sampled for virus titration at 5, 7 and 10 days post-transfection. To confirm the presence of the mutations in 3'UTRs, viral RNA was isolated from culture fluid samples using QIAamp Viral RNA Kit (Qiagen, Germany) and used as a template for RT-PCR with 3'UTR sequencing primers (Supplementary Table 2). RT-PCR was performed using SuperScript III One-Step RT-PCR Kit with Platinum Taq (Invitrogen, USA) and the following cycling conditions: 15 min at 60 °C, 2 min at 95 °C; 35 cycles of 15 s at 95 °C, 30 s at 55 °C, 40 sec at 68 °C; and a final extension for 5 min at 68 °C. PCR products were separated in 2% agarose gel, purified using QIAquick Spin Gel extraction kit (Qiagen, Germany) and Sanger Sequenced.

**Virus growth kinetics**. The growth kinetics of wild type and mutant viruses were assessed in RML-12 cells. Cells were seeded at $2 \times 10^6$ cells per well in 6-well plates and inoculated with wild type or mutated viruses at an MOI of 0.1 by incubating for 1 h with 200 μL of virus inoculum. Incubations were performed at 28 °C. Inoculum was then removed, and cells were washed three times with PBS and overlaid with 2 mL of L-15 culture medium supplemented with 2% FCS. At time point zero, 100 μL of media was immediately collected, and infected cells were then incubated for 7 days at 28 °C. Culture fluid samples (100 μL) were then harvested at 3, 5 and 7 days post-infection and subjected to foci-forming immunoassay to determine the virus titres, from which growth curves were plotted.

**Statistical analysis**. Statistical analyses were performed using GraphPad Prism v.9.0. Exact statistical tests are specified in figure legends.

**Reporting summary**. Further information on research design is available in the Nature Research Reporting Summary linked to this article.

## Data availability
The SHAPE data generated in this study have been deposited in the Figshare database under accession code 19100390. The raw gel quantification and virus titration data generated in this study are provided in the Source Data file. All other data are available within the paper and in supplementary materials. Source data are provided with this paper.

## Code availability
The study did not utilise any custom codes. Source data are provided with this paper.

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

## Acknowledgements

We are grateful to Prof Jeffrey S. Kieft (University of Colorado) for providing key advice on the structural aspects of the study. Work was funded by the Australian Research Council (ARC) grant DP190103304 to AAK, RAH, AS, and JHP and Medical Research Council (MRC) grant MR/N01054X/1 to AT. Sanger sequencing was performed by the Australian Genomics Research Facility (Brisbane, Australia).

## Author contributions

A.S., A.A.K. – conceptualization; A.S. – experiment design, A.S., B.P., J.D.J.S., X.W., T.B., F.J.T., A.T. – experiments; A.S., R.P. – data analysis, bioinformatics; J.H., A.M.G.C., J.H.P., R.A.H. – critical materials and reagents; A.A.K., A.T. – project supervision; A.A.K., A.S., R.A.H., J.H.P., A.T. – funding acquisition.

## Competing interests

The authors declare no competing interests.
