## [Peer Review File · Nature Communications]

Peer reviewer comments, first round

Reviewer #1 (Remarks to the Author):

This study demonstrates that a variety of insect-only flaviviruses contain structural elements in their 3' UTRs that stall XRN1 5'-3' exonucleolytic decay and generate sfRNAs both in biochemical assays as well as in virus-infected cells. Furthermore, the authors use SHAPE analysis to explore the base-pairing of residues in these XRN1-resistant structures (xrRNAs) and characterize/classify the various structures that were uncovered in the context of previous work on xrRNAs in the field. Finally, individual xrRNA elements are mutated in a variety of insect-specific flaviviruses to establish a requirement for at least one functional 3' UTR element for efficient viral replication. Overall the data are robust and largely support the conclusions that are drawn. This study complements the extensive work that has already been published on sf/xrRNA structure/function. Thus it does not necessarily break a great deal of novel ground from a mechanistic perspective, but it will be a welcomed addition to specialists in the field. I do have a few suggestions to polish the study/manuscript:

Major Point:

1. Fig. 2D/E: While the authors conclude that the nPK elements that they have uncovered are 'xrRNAs', it is not clear to this reviewer whether they represent truly independent XRN1-resistant elements or are simply accessory structures that assist in the formation/stabilization (directly or perhaps indirectly) of the bona fide class 1a ZIKV-like xrRNA elements that preceded them. Thus the authors should either temper their conclusion/clarify this section or directly test the ability of the nPK elements to function independently.

Minor Points:

1. Fig. 1B: the extent of Xrn1 KD that was achieved in these experiments should be reported.
2. Fig. 2: The legend for panel E states that it's a quantitation of panel C rather than panel D
3. Line 219: I believe that the authors mean Fig. 2D rather than Fig 1D
4. The manuscript as currently presented is very acronym-heavy and will likely be very difficult for a general, non-specialist reader to digest. I do not have any specific recommendations for revision, but wanted to point this out as it could limit the broader impact of the work.

Reviewer #2 (Remarks to the Author):

These authors demonstrate an interesting role of sfRNA in ISFs. Firstly, Northern blotting shows that sfRNA was produced in several ISF infected cells and this production is mediated by XRN1-resistant mechanism. Additional structural probing reveals important elements that confer XRN1 resistance. Also, these authors performed sequence- and structure-based phylogenetic analysis of xrRNAs and 3'UTRs of several flaviviruses. Lastly, functional significance of xrRNA (and sfRNA) was tested using various mutant viruses. Overall, this study is interesting and would expand our current knowledge of sfRNA biogenesis in flaviviruses, particularly from the perspective of RNA structure. As these structural probing data were acquired from IVT RNA in test tubes, it would be of interest to see if similar structures would also form in infected cells. Additional points suggested to clarify are listed below.

- Authors indicate that flavivirus KRBV produced only one sfRNA (Fig 1A) and only one sfRNA was observed in Fig 4C. However, there are multiple bands in Fig 1A. Is it possible that more than one species of sfRNA is produced in KRBV infected cells?

- In supplementary Fig 2, it would be more informative to provide the proportion of each sfRNA from the sequencing results? How many sequencing results show sfRNA1 and how many results for sfRNA2? Presumably, this proportion should agree with band intensity observed in Northern blots?

- In Fig 1B, it would be helpful to perform densitometry analysis to quantify band intensity of each sfRNA upon XRN1 knock down. In addition, the same analysis could be done for Fig 3C and 3G as it has been done in Fig 2D and 2E.
- Does BinJV produce sfRNA3? In Fig 1A, a band close to 200bp is indicated as sfRNA3. However, this sfRNA3 does not seem to be affected by XRN1 knock-down in Fig 1B. And in Fig 2D no sfRNA3 of BinJV was observed in vitro. Please clarify.
- SHAPE reactivity provides information about RNA structure stability and energetics. Does this correlate with XRN1 resistance and the abundance of each sfRNA?
- It is interesting that PaRV deficient in major sfRNA1 and 3 replicates less efficiently as compared with WT whereas sfRNA2 deficiency has little effect on viral fitness (Fig 6B). Can the replication defect of PaRV sfRNA1 and sfRNA3 mutants be rescued by complementing sfRNA1 or sfRNA3 in trans?
- Line 629 C6/C6 cells – typo, should be C6/36

Reviewer #3 (Remarks to the Author):

This manuscript describes a study of the structures, functions and evolution of the XRN-1 blocking elements (xrRNA) in the 3' UTRs of the RNA genomes of the known members of the different clades of the genus Flavivirus. The sfRNA producing XRN-1 elements of pathogenic, dual host, mosquito borne and tick borne flaviviruses have been well studied previously but only one of these elements has been studied in a genome of a nonpathogenic flavivirus that either infects only mosquitoes or infects mammals but has no known vector. SHAPE analyses were used to obtain structural data on possible xrRNA elements in the genomes of several mosquito-specific flaviviruses and the ability of each xrRNA element to inhibit the progression of XRN-1 in vitro and generate sfRNAs in infected cells was analyzed. The importance of each sfRNA for enhancing virus replication was assessed by testing mutated xrRNAs. Both the 3' UTR sequence and xrRNA element structures were used for a phylogenetic analysis that identified a near common ancestor with a single "hybrid" xrRNA element. Multiple independent insertion events of one or more additional xrRNA elements occurred during evolution of the flavivirus clades. Additional classes of known xrRNA elements as well as a novel xrRNA element were identified. The data presented are convincing and support the authors conclusions. A previous study showed that one of the two sfRNAs produced by DENV 2 functioned during replication in the mammalian host and the other functioned in the mosquito host suggesting that xrRNA duplication is an adaptation to host switching. However, the data in this manuscript indicate that the multiple xrRNAs in the genomes analyzed provide redundant "backup" function in the same host. The congruence of the overall topology of the polyprotein and 3' UTR phylogenetic trees indicated co-evolution between these genome regions. Overall, the results presented in this manuscript represent a significant advance in knowledge in the field. The manuscript is clearly written but could be improved by some additional editing.

Concerns/suggested edits:

A statement about the discovery of a novel structure (n-xrRNA) could be included in the abstract. Starting with the introduction (L76), abbreviations are often not spelled out the first time they are used in the text. This is especially true of the virus names. A supplementary table containing a list of all of the full virus names and their abbreviations should be added.

L39-41- The logic of this statement is not clear. How would multiple xrRNA be selected to protect these structures from harmful mutations? It seems that additional structures would be generated by insertions and this would ensure maintenance of the function through redundancy.

"for the first time" should be deleted from multiple statements in the Results.

L81- ...MBF (WNV)...

L82- This pseudoknot has 6 not 5 nts

There are many instances where "the" or "a" is omitted.

L95- ...and switches...

L118- ...should allow elucidation of whether ...

It does not appear that the mutation strategy and specific nts changed to disable the different xrRNAs, are described but I may have missed this.

L290-292- The meaning of this sentence is not clear as written.

L434-435- Structures with unique functions would not necessarily equally impair virus replication.

L457- For PaRv, for which...

L463 and 465- Did this virus revert by day 7?

L471- The initial low titer is not obvious on the graph (Fig. 6A)

L473-476- Additional information is needed here to explain what the mutations were and why they had a low effect.

The first two paragraphs of the discussion could be combined to reduce redundancy.

L537- What is TABV?

RESPONSE TO REVIEWERS COMMENTS

Dear reviewers,

Thank you for the comments on our manuscript. We have taken all of the comments into consideration and revised the manuscript accordingly. We believe that additional experiments and analyses performed in revision together with modifications of the text have substantially improved the manuscript and resolved all concerns raised in peer-review. Please find our detailed response to your comments below.

Reviewer #1 (Remarks to the Author):

Major Point:

1. Fig. 2D/E: While the authors conclude that the nPK elements that they have uncovered are 'xrRNAs', it is not clear to this reviewer whether they represent truly independent XRN1-resistant elements or are simply accessory structures that assist in the formation/stabilization (directly or perhaps indirectly) of the bona fide class 1a ZIKV-like xrRNA elements that preceded them. Thus the authors should either temper their conclusion/clarify this section or directly test the ability of the nPK elements to function independently.

We thank the reviewer for raising this important point. To address this, we performed *in vitro* XRN-1 resistance assay with GFP RNA fragments bearing an insertion of nPK elements or canonical xrRNA. We found that nPKs cannot resist XRN-1 activity on their own (Supplementary Figure 4). Taking into account the results of mutational analysis of PKs and nPKs which showed that mutations in canonical PKs did not completely abolish XRN1 resistance while mutations in nPKs retained partial XRN1 resistance (Fig 2D,E) we now conclude that nPKs likely act by stabilizing the adjacent canonical xrRNAs to ensure their complete XRN1 resistance. We have modified the text of the manuscript accordingly by removing our statements that nPKs represent independent "xrRNAs" and by clarifying their auxiliary role in providing additional XRN1 resistance to the upstream canonical xrRNAs.

Minor Points:

1. Fig. 1B: the extent of Xrn1 KD that was achieved in these experiments should be reported.

We have incorporated Supplementary Figure 2B into the revised manuscript, which shows that knock-down efficiency of 90-98% was achieved as indicated by qRT-PCR for XRN-1 mRNA.

2. Fig. 2: The legend for panel E states that it's a quantitation of panel C rather than panel D

Corrected in the revised manuscript.

3. Line 219: I believe that the authors mean Fig. 2D rather than Fig 1D

Corrected in the revised manuscript.

4. The manuscript as currently presented is very acronym-heavy and will likely be very difficult for a general, non-specialist reader to digest. I do not have any specific recommendations for revision, but wanted to point this out as it could limit the broader impact of the work.

In the revised manuscript we spelled out previously abbreviated terms wherever practical

Reviewer #2 (Remarks to the Author):

As these structural probing data were acquired from IVT RNA in test tubes, it would be of interest to see

if similar structures would also form in infected cells.

Although the method for *in cellulo* SHAPE does exist, it cannot be applied to determine the structure of viral 3'UTRs in flavivirus infected cells as the infected cells contain multiple forms of 3'UTR-containing RNAs, including circularised form of viral genomic RNA and double stranded RNA replicative intermediate, as well as sfRNA- and viral RNA-protein complexes. They will all produce overlaying SHAPE signals due to different nucleotide pairing in each form of RNA, thus yielding the data that cannot be interpreted. Therefore, it is technically impossible to identify the structure of ISF 3'UTRs in infected cells. In addition, to mimic the biochemical conditions in the cells, we performed SHAPE at 28C and used physiological concentrations of salt and magnesium. Therefore, we believe that identified structures mimic closely the structures formed by the 3'UTR of linear form of viral genomic RNA in in infected cells.

- Authors indicate that flavivirus KRBV produced only one sfRNA (Fig 1A) and only one sfRNA was observed in Fig 4C. However, there are multiple bands in Fig 1A. Is it possible that more than one species of sfRNA is produced in KRBV infected cells?

The presence of the additional bands was inconsistent between individual mosquitoes and also occurred in Mock in some instances. Therefore, we concluded that they resulted from the nonspecific probe binding. In revision we have optimised the Northern Blotting method for the detection of KRBV sfRNA. The new blot, which replaces the original panel in Fig 1A, does not contain these bands.

We also provide below the uncropped image of the blot for the reviewer consideration on which RNA from three KRBV-positive (KRBV lanes) and three KRBV-negative (Mock lanes) mosquitoes was probed for KRBV sfRNA. The blot shows that the larger band considered as KRBV sfRNA in the manuscript is the only one that consistently occurs in KRBV – positive mosquitoes, and not in uninfected ones. Therefore, we are convinced that KRBV produces only one sfRNA isoform.

- In supplementary Fig 2, it would be more informative to provide the proportion of each sfRNA from the sequencing results? How many sequencing results show sfRNA1 and how many results for sfRNA2? Presumably, this proportion should agree with band intensity observed in Northern blots?

As sequencing for Supplementary Figure 2 was performed using Sanger method on gel-extracted individual PCR products for each sfRNA, it is impossible to estimate the relative abundance of individual sfRNAs based on these data. However, to address the reviewer's comment, we performed densitometry for each individual sfRNA band detected in Northern blots and included these data into the revised manuscript as Supplementary Figure 2A.

- In Fig 1B, it would be helpful to perform densitometry analysis to quantify band intensity of each sfRNA upon XRN1 knock down. In addition, the same analysis could be done for Fig 3C and 3G as it has been done in Fig 2D and 2E.

In revision we performed the suggested densitometry analyses on the blots showing sfrRNA production in XRN1-depleted cells and presented them as a supplementary Figure 2C of the revised manuscript. Unlike Figs 2D and E, the Figs 3C and 3G showed that mutations in pseudoknots eliminated production of the respective sfrRNAs leaving no residual bands for densitometry analysis. Therefore, densitometry for these figures was not performed.

- Does BinJV produce sfrRNA3? In Fig 1A, a band close to 200bp is indicated as sfrRNA3. However, this sfrRNA3 does not seem to be affected by XRN1 knock-down in Fig 1B. And in Fig 2D no sfrRNA3 of BinJV was observed *in vitro*. Please clarify.

Production of sfrRNAs from dumbbells was previously reported to be XRN-1-independent, although the exact nuclease responsible for it is yet to be identified (ref 26 in the manuscript). Our results for BinJV are consistent with this report. We mentioned this in the introduction of the original manuscript (lines 90-92 of original manuscript; 89-92 of revised manuscript). We have now added more explanation and reference to the corresponding section in the Results (lines 184-187, revised manuscript).

- SHAPE reactivity provides information about RNA structure stability and energetics. Does this correlate with XRN1 resistance and the abundance of each sfrRNA?

SHAPE reactivity only provides information about pairing status for each individual nucleotide. Free energy of the secondary structure can be predicted based on SHAPE-guided modelling. However, this information is not sufficient to calculate free energy of the tertiary structure which is responsible for XRN1-resistance. Solving the crystal structures is the only way to determine free energies of the XRN-1 resistant ring-like folds. Accordingly, we cannot conclusively correlate xrRNAs stability with the abundance of each corresponding sfrRNA based on our SHAPE data.

- It is interesting that PaRV deficient in major sfrRNA1 and 3 replicates less efficiently as compared with WT whereas sfrRNA2 deficiency has little effect on viral fitness (Fig 6B). Can the replication defect of PaRV sfrRNA1 and sfrRNA3 mutants be rescued by complementing sfrRNA1 or sfrRNA3 *in trans*?

The reason that sfrRNA2 deficiency had little effect on PARV virus replication is likely due to suboptimal structure of xrRNA2 that leads to weak XRN1 resistance and consequently low abundance of sfrRNA2 in WT virus infection. This is explained in detail in the Results section describing data in Fig 2A-C and Fig 6B.

While *trans*-complementation of sfrRNA1- and sfrRNA3-deficient virus mutants with sfrRNA1 and sfrRNA3 would provide additional confirmation for the role of these sfrRNAs in virus replication, these experiments are not straightforward. Firstly, sfrRNA-1 if provided *in trans* will be processed by cellular XRN-1 into both smaller species: sfrRNA2 and sfrRNA3. Secondly, our attempts to transfect *in vitro* transcribed sfrRNAs into insect cells resulted in extensive cell death, likely due to the ability of the sfrRNAs to induce cytopathic effect previously described for vertebrate cells.

- Line 629 C6/C6 cells – typo, should be C6/36

This has been corrected in the revised manuscript.

Reviewer #3 (Remarks to the Author):

Concerns/suggested edits:

A statement about the discovery of a novel structure (n-xrRNA) could be included in the abstract.

Based on our new results generated in response to Reviewer 1 concern, we added the following statement into the abstract of the revised manuscript:

We demonstrate that 3'UTRs of all classical ISFs, except Anopheles spp-associated viruses, and of the dual-host associated ISF Binjari virus contain duplicated xrRNAs. We also reveal novel structural elements in the 3'UTRs of dual host-associated and Anopheles-associated classical ISFs.

Starting with the introduction (L76), abbreviations are often not spelled out the first time they are used in the text. This is especially true of the virus names. A supplementary table containing a list of all of the full virus names and their abbreviations should be added.

The original manuscript contains Supplementary table 1, which spells out the abbreviations of all viruses mentioned in the study. In revision we also provided full virus names in the text of the manuscript where they are used for the first time.

L39-41- The logic of this statement is not clear. How would multiple xrRNA be selected to protect these structures from harmful mutations? It seems that additional structures would be generated by insertions and this would ensure maintenance of the function through redundancy.

This statement has been reworded as follows:

Our data thus provide evidence that duplicated xrRNAs are selected in the evolution of flaviviruses to provide functional redundancy which preserves the production of sRNA if one of the structures is disabled by mutations or misfolding.

“for the first time” should be deleted from multiple statements in the Results.

In the revised manuscript we have deleted “for the first time” from all the statements in the Results section.

L81- ...MBF (WNV)...

Spelled out in the revised manuscript

L82- This pseudoknot has 6 not 5 nts

Corrected in the revised manuscript

There are many instances where “the” or “a” is omitted.

The revised manuscript has undergone additional proofreading to correct this

L95- ...and switches...

L118- ...should allow elucidation of whether ...

Corrected in the revised manuscript

It does not appear that the mutation strategy and specific nts changed to disable the different xrRNAs, are described but I may have missed this.

The mutagenesis strategy is described in “Methods” section, and specific nucleotide substitutions for PaRV, PCV and KRBV are specified in the legend for figures 3 and 4. In the revised manuscript we also added specific nucleotide changes for BinJV mutants in the legend for Fig 2.

L290-292- The meaning of this sentence is not clear as written.

In the revised manuscript we split the sentence and now it reads as follows:
Based on SHAPE-guided folding of PCV 3'UTR we identified three copies of the structural element which has the features of a potential xrRNA. Each of them consisted of a PK-forming SL followed by two small SLs. At the 3'-end PCV 3'UTR contained two short hairpins and a 3'SL (Fig 3E).

L434-435- Structures with unique functions would not necessarily equally impair virus replication.

We deleted "equally" and now this statement reads as follows:
If individual sRNA species have unique, specialised functions, we expected that loss of each sRNA should impair virus replication.

L457- For PaRv, for which...

Corrected

L463 and 465- Did this virus revert by day 7?

We performed Sanger Sequencing of all mutant viruses used in this experiment at the final time point (7dpi) and only found reversion in PCV PK1'2'3' mutant. As the northern blotting in Fig 6A was performed at the matching time point and showed that all viruses except PCV PK1'2'3' maintained the expected phenotype of sRNA production, indicating that they did not revert, we did not mention the sequencing results in the original manuscript. We added the statement in the revised manuscript, which declares that reversion did not occur.

L471- The initial low titer is not obvious on the graph (Fig. 6A)

This statement refers to the titer of P0 virus, which are not shown in the manuscript. Fig 6A at time point 0 indicates the amount of the residual inoculum left in the wells after infection and washing. As infection with all viruses was performed at the same MOI, the 0dpi titers in Fig 6A are similar.

L473-476- Additional information is needed here to explain what the mutations were and why they had a low effect.

The text in the revised version now reads:

PK1', PK2' and PK1'2' mutants of BinJV also replicated at the levels comparable to the WT virus at all time points (Fig 6B). However, these mutant viruses were not fully deficient in either sRNA-1 or sRNA-2 (Fig 6A), consistent with the in vitro XRN1 digestion assay (Fig 2D.E). This further demonstrates the functional significance of novel pseudoknots for enabling the production of sRNAs (albeit at reduced levels) when xrRNA1 and xrRNA2 are affected by mutations (Fig 6A).

The first two paragraphs of the discussion could be combined to reduce redundancy.

We have combined the first two paragraphs of the discussion in the revised manuscript.

L537- What is TABV?

TABV is Tamana Bat Virus

Peer reviewer comments, second round

Reviewer #1 (Remarks to the Author):

The authors have done a thorough job in addressing the comments raised in the initial round of review. I find the revised manuscript to be improved, convincing and an interesting contribution to the field

Reviewer #2 (Remarks to the Author):

The authors have addressed my concerns. I do not have additional questions at this point.

Reviewer #3 (Remarks to the Author):

The authors have adequately addressed this reviewer's concerns in the revised manuscript. I do not have any additional comments.